# TYGS is an automated high-throughput platform for state-of-the-art genome-based taxonomy

Jan P. Meier-Kolthoff [1] & Markus Göker [1]

Microbial taxonomy is increasingly influenced by genome-based computational methods. Yet such analyses can be complex and require expert knowledge. Here we introduce TYGS, the Type (Strain) Genome Server, a user-friendly high-throughput web server for genome-based prokaryote taxonomy, connected to a large, continuously growing database of genomic, taxonomic and nomenclatural information. It infers genome-scale phylogenies and state-of-the-art estimates for species and subspecies boundaries from user-defined and automatically determined closest type genome sequences. TYGS also provides comprehensive access to nomenclature, synonymy and associated taxonomic literature. Clinically important examples demonstrate how TYGS can yield new insights into microbial classification, such as evidence for a species-level separation of previously proposed subspecies of *Salmonella enterica*. TYGS is an integrated approach for the classification of microbes that unlocks novel scientific approaches to microbiologists worldwide and is particularly helpful for the rapidly expanding field of genome-based taxonomic descriptions of new genera, species or subspecies.

[1] Leibniz Institute DSMZ—German Collection of Microorganisms and Cell Cultures, Inhoffenstraße 7B, 38124 Braunschweig, Germany. Correspondence and requests for materials should be addressed to J.P.M.-K. (email: jan.meier-kolthoff@dsmz.de)

Earth is dominated by microbes as these regulate and sustain life in various ways[1,2]. They exceed all other groups of organisms with regard to their number of individuals, biomass, and genetic diversity[1,3]. As countless applications in medicine, biotechnology, and ecology heavily rely on a most accurate understanding of this tremendous microbial variety, the classification, characterization, and identification of *Bacteria* and *Archaea* is of utmost importance.

Since the late 19th century, technological advances have continuously influenced taxonomic procedures and techniques[4], ranging from analyses on morphology, physiology and biochemistry, chemotaxonomy, numerical taxonomy, DNA G+C content, DNA:DNA hybridization (DDH), and DNA:rRNA hybridization over genotypic analyses (e.g., 16S rRNA) to analyses of whole genomes. Even though these methods can contribute toward the taxonomic positioning of samples, bacterial taxonomy is nowadays increasingly influenced by genome-based approaches[5]. Whereas conventional DDH and G+C content measurement are known[6,7] to be rather error-prone, tedious and only available in specialized laboratories, their genome-based counterparts provide a new level of standardization due to instant reproducibility and validation, yielding a huge gain in time and methodological accuracy[7].

For example, within the realm of in silico species delineation methods[8,9], digital DDH (dDDH) outperformed average nucleotide identity (ANI)[6,7], can be utilized for taxon delineation at the subspecific level[10] and benefits from optimizations of the underlying Genome BLAST Distance Phylogeny method (GBDP) for phylogenomic analyses[7,11]. Similarly, the G+C content is an important taxonomic marker in the genomic era, indicating affiliations to distinct species if the difference between two genome sequences exceeds 1%[6].

Phylogenetic analyses of the 16S rRNA gene sequence[4] nowadays still heavily contribute to our understanding of prokaryotic systematics, but these are incapable of unambiguously resolving evolutionary relationships within many groups[4]. Although less frequently, the problem can also arise in multilocus sequence analyses (MLSA), which use a larger set of loci. However, an approach that uses only a fraction of the genes can hardly be considered a true genome-scale phylogenetic method[12]. To that end, truly whole-genome-based methods such as GBDP have been developed[7,11,13,14] to provide unprecedented insights into the microbial tree of life[6,10,15–21], to elucidate evolutionary relationships of viruses and eukaryotes[22,23] and to yield robust branch support values[5,24,25].

Type strains form the backbone of prokaryotic systematics as nomenclatural types of species and subspecies[26], and comparisons with established type strains are mandatory when classifying novel strains[4]. Although the cultivation of many groups of microbes is notoriously difficult[27], the challenges for switching to an approach in which, e.g., genome sequences could act as type material, must not be underestimated[28]. In fact, the number of species names validly published per year has steadily increased during the last decades—even if one disregards all new combinations for existing species names—yielding > 900 in 2018, and the vast majority of these taxa are novel species in existing genera (Supplementary Fig. 1). Particularly, the number of species names proposed by authors from China and Southeast Asia is growing rapidly[29].

As the acceptance and practical relevance of genome-based taxonomy[8,9] currently depends on the availability of type strain genomes, large-genome sequencing projects such as the Genomic Encyclopedia of *Bacteria* and *Archaea* (GEBA) initiative significantly expanded the genomic coverage of the microbial tree of life[15]. In addition, the International Journal of Systematic and Evolutionary Microbiology, in which most descriptions of new species are published, recently made it mandatory for such descriptions to be accompanied by a genome sequence (http://ijs. microbiologyresearch.org/content/Genome_data_required_IJSEM. Accessed 22 April 2019). However, in order to meet the apparent huge demand for reliable, genome-based taxonomic methods, publicly available state-of-the-art platforms are necessary that are easy to use and free of charge.

Yet two major problems still remain. First, if a type (strain) genome sequence is publicly available, it might be difficult to verify its origin, especially in a complex situation of different synonyms and strain deposits. Second, taxonomists are not necessarily trained in bioinformatics and cannot necessarily afford organizing and maintaining a computing infrastructure. The integration of a database of type (strain) genomes and phylogenomic methods is thus a major step forward. Having both data validation and methods integrated in a single platform is clearly synergetic: relevant type (strain) genomes can be automatically determined and various downstream analyses can be conducted automatically in a high-throughput approach, without any manual interaction.

Yet established platforms only cover certain aspects of this concept and have in common that they do not represent truly genome-based frameworks for taxonomic classification and identification. For example, EzBioCloud is a mixed database containing 16S rRNA gene and genome sequences (where available) as well as of type strains and non-type strains, but users can only upload 16S rRNA gene sequences for identification[30]. Sequences are not phylogenetically analyzed and only subjected to a comparison against an internal 16S rRNA gene sequence database. The actual identification process is done against an underlying artificial taxonomic backbone that uses genome sequences primarily for species delineation in edge cases[30]. Whereas the JSpeciesWS web server allows for the upload of a small number of genome sequences, it only reports different types of pairwise ANI and Tetra values[31]. Even though JSpeciesWS also offers a manual selection of (non-)type strains, the underlying database (Ensembl Bacteria) uses the GenBank classification, which is not an authoritative source of taxonomy and nomenclature[32]. Moreover, the type strains closest to the query strain are not automatically detected.

One of the primary purposes of microbial genomics is a global understanding of the microbial world, by exploring and understanding microbial diversity and elucidating its evolutionary dynamics[2]. TYGS (https://tygs.dsmz.de), the Type (Strain) Genome Server, contributes toward that goal by providing an integrated approach on genome-based taxonomy by joining features such as a comprehensive database of the genomes of type strains of species and subspecies with validly published names, automated detection of closest neighbors of query genomes, and truly whole-genome-based methods for phylogeny and classification. These replace taxonomic standard techniques such as DDH, G+C content and 16S rRNA gene and MLSA, yielding a unique combination of tools to be explored by microbiologists, taxonomists, and health professionals alike.

## Results

**Overall TYGS workflow.** The TYGS workflow is shown in Fig. 1, whereas an in-depth description is found in the Methods chapter. By incorporating the techniques of the Genome-to-Genome Distance Calculator[7], TYGS compares user genomes against its database of type (strain) genomes, followed by inference of phylogenetic trees (instead of mere clusterings) with branch support and an indicator of treelikeness[13,33], classification at the species and subspecies level, reports of the differences in genomic G+C content[6] and all relevant taxonomic literature for

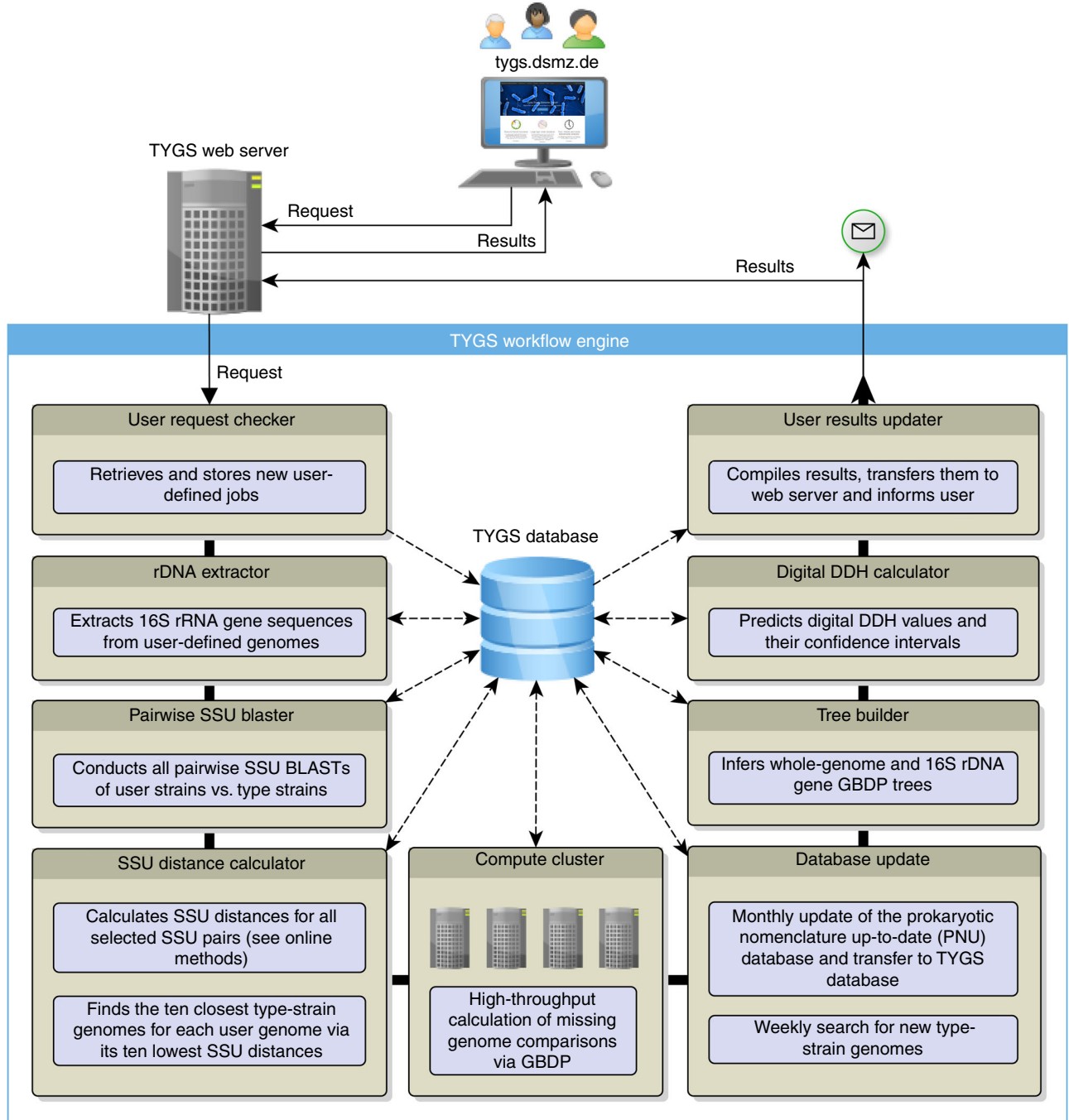

**Fig. 1** TYGS workflow. First, a request is prepared and submitted by a user to the TYGS web server. The TYGS workflow engine periodically checks for new user requests and imports these data into the central TYGS database. The data are processed by several independent services. Once the results arrive at the web server the user gets informed via e-mail and can conveniently access the results via the web browser of choice. Solid lines indicate the logical program flow, whereas data flow from and to the database is indicated by dashed lines

download. Users can upload wholly or partially finished genome sequences; TYGS will determine the 10 most closely related type strains for each user genome by default. Assignment to species and subspecies is based on the established dDDH thresholds[7,10] and a clustering algorithm specifically designed for taxonomic conservatism and to take nomenclatural priority into account. The trees and their annotation can be explored and downloaded via an interactive viewer.

**Exemplary analyses of two clinically important data sets**. To demonstrate the capabilities of TYGS, we analyzed two data sets

of clinically important[34,35] bacteria from the genera *Myco-bacterium* and *Salmonella*. A platform such as the TYGS is expected to be able to confirm results based on modern genome-based taxonomic approaches but to also yield new insights into the classification of pathogens as well as any other kinds of *Bacteria* and *Archaea*.

Species from the *Mycobacterium tuberculosis* complex (MTBC) caused c. 9 million diseases and c. 1.5 million deaths in 2013[36]. Numerous taxonomic changes within the MTBC took place over time, culminating in recent studies using genome sequencing[37]. The TYGS result shown in Fig. 2 revealed a highly supported

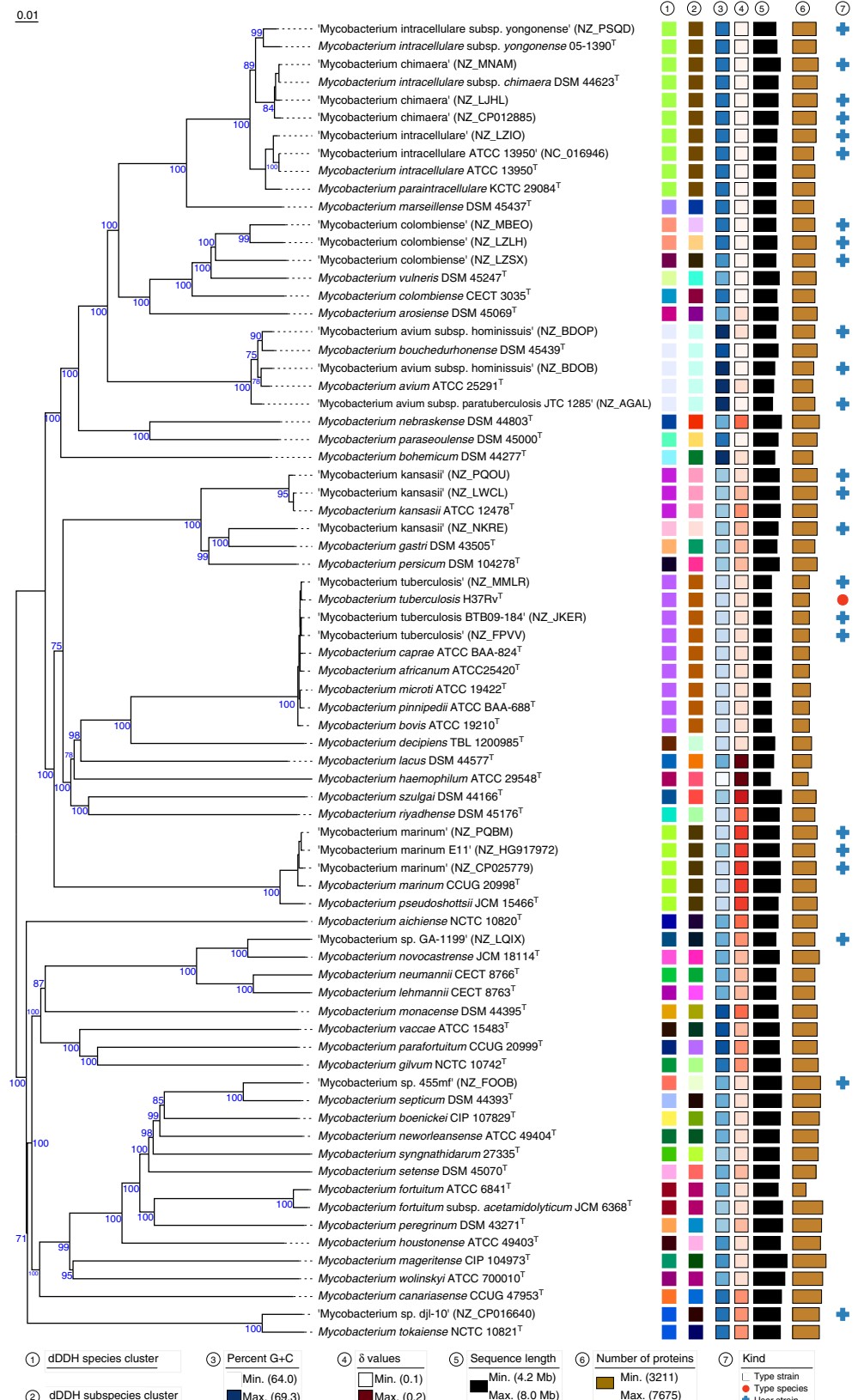

**Fig. 2** TYGS result for the *Mycobacterium* data set. Tree inferred with FastME 2.1.4[56] from GBDP distances calculated from genome sequences. Branch lengths are scaled in terms of GBDP distance formula $d_5$; numbers above branches are GBDP pseudo-bootstrap support values from 100 replications. Leaf labels are annotated by affiliation to species (1) and subspecies (2) clusters, genomic G+C content (3), $\delta$ values (4), overall genome sequence length (5), number of proteins (6), and the kind of strain (7). User-provided GenBank accession IDs are shown in parentheses; master record accessions are truncated

clade corresponding to a species cluster representing the MTBC, including the type strains of *Mycobacterium tuberculosis*, *M. bovis*, *M. africanum*, *M. pinnipedii*, *M. caprae*, and *M. microti*. This implies that the other MTBC species have to be regarded as later heterotypic synonyms of *M. tuberculosis* because they are even identical to it at the subspecies level, which is in full agreement with the most recent taxonomic study[37]. Type strains of relevance were automatically detected by the TYGS for clarifying taxonomic questions associated with each user-defined genome sequence. The overall treelikeness of the data set appeared to be high (as revealed by the low δ values[13,33]), corresponding to high branch support on average. The extended TYGS 16S rRNA gene analysis (Supplementary Fig. 2, Supplementary Data 1), which also covers type strains not yet represented by a genome sequence, would tell the user that most additional type strains of other species with validly published names are rather not expected to have to be regarded as later heterotypic synonyms of *M. tuberculosis*. Only *M. shottsii* and *M. ulcerans* showed a 16S rRNA gene sequence similarity to *M. tuberculosis* slightly above the threshold for the phylum Actinobacteria that requires an additional DDH analysis to clarify the species status[38]; so far only some of the necessary comparisons were made and only using conventional DDH[39].

*Salmonella* is a major cause of foodborne illness throughout the world. Infections by nontyphoidal *Salmonella* strains alone are estimated to result in ~ 155,000 deaths per year[34]. The estimated number of *Salmonella* species varied considerably over time[40]. Although the "one serotype–one species" approach would have resulted in > 2000 *Salmonella* species[40] it was also proposed to consider each of the previously recognized four subgenera as a single species[41] or to place all serotypes known at that time into a single species, based on conventional DDH experiments[42]. The most-recent taxonomic view accepts three species, *Salmonella bongori*, *S. subterranea*, and *S. enterica*, as well as the six subspecies *S. enterica* subsp. *arizonae*, *S. enterica* subsp. *diarizonae*, *S. enterica* subsp. *enterica*, *S. enterica* subsp. *houtenae*, *S. enterica* subsp. *indica*, and *S. enterica* subsp. *salamae*, whereas the remaining species are regarded as later heterotypic synonyms of *S. enterica*[40,43].

The TYGS analysis of the *Salmonella* data set (Fig. 3) revealed a maximally supported *Salmonella* subtree to the exclusion of *Salmonella subterranea*. This species was already shown to phylogenetically not belong to *Salmonella*[44] but accordingly proposed new names where not validly published yet. The example illustrates how the TYGS helps elucidating not only the boundaries of species and subspecies but also those of genera, based on the criteria of monophyly and taxonomic conservatism[5,20,25,45].

More interestingly, *S. enterica* was split into six well-supported species clusters corresponding to reasonably supported clades (Fig. 3), indicating that most of the currently recognized subspecies[40,43] should be elevated to species status. Only in the case of *S. enterica* subsp. *salamae* care must be taken because the type strain is not represented in the data set, whereas both an *S. enterica* subsp. *indica* and an *S. enterica* subsp. *diarizonae* genome sequence from GenBank were apparently mislabelled. The extended TYGS 16S rRNA gene analysis (Supplementary Fig. 3, Supplementary Data 2), which also covers type strains not yet represented by a genome sequence, would reveal that *S. enterica* subsp. *salamae* also belongs to the clade even if other strains supposed to belong to the subspecies were not present in the data set.

The inter- and intra-subspecies distribution of the dDDH values revealed pronounced offsets surrounding the 70% species delineation threshold (Supplementary Fig. 4, Supplementary Data 3), leaving little doubt that most of the subspecies are indeed separated at the species level. Whereas this outcome is not unexpected given previous results on sexual isolation between the subspecies[46], to the best of our knowledge a taxonomic assessment of the group on genome-based methods that correspond to the gold standard for species delineation[8,9] has not yet been carried through before. In fact, a separate species with a validly published name was as yet only proposed for one of the currently supposed subspecies of *S. enterica*, *S. enterica* subsp. *arizonae*. The analysis did nevertheless also confirm conclusions well known from the literature, such as the status of *S. choleraesuis*, *S. enteritidis*, and *S. typhimurium* as heterotypic synonyms of *S. enterica*. High treelikeness (low δ values) and resolution (branch support) appeared to be comparable to the *Mycobacterium* analysis except for the (expected) fading of the phylogenetic signal within the clades corresponding to the subspecies clusters.

These results can be interactively explored on the TYGS web page (https://tygs.dsmz.de), which also provides other example data sets as well as a tutorial.

## Discussion

The analysis of data sets from distinct bacterial genera demonstrated how TYGS can be used to rapidly elucidate the taxonomic situation within a particular group of organisms and how new insights can be obtained. The results include a genome-scale phylogeny with branch support values and treelikeness indicators; estimates for species and subspecies boundaries; names, synonyms, and authorities of the taxa involved; links to deposits of existing type strains in culture collections; links to the taxonomic literature that is available online; download of all taxonomic literature relevant for the investigated organisms in BibTeX format; links to the BacDive database[47] for obtaining additional information on the selected species or subspecies such as their higher classification; genome statistics such as G+C content, genome size, and number of proteins; and annotated trees that can be explored interactively.

The focus of the TYGS on the types of species and subspecies with validly published names ensures that only taxonomically relevant strains are considered[26]. Among the currently 15,898 species and subspecies with validly published names (18,472 names including synonyms) in the DSMZ Prokaryotic Nomenclature Up-to-date (PNU) database > 8000 type strains are genome-sequenced and covered by the TYGS database as of November 2018. The missing taxon names are precisely those that are not validly published and cannot obtain priority over as yet validly published names. For instance, a name in the category *Candidatus* cannot be validly published, a restriction not imposed by the TYGS but by the International Code of Nomenclature of Prokaryotes[26], which does not cover the category *Candidatus*. However, users are, of course, free to add genome sequences of *Candidatus* taxa to their TYGS analysis, much like any other kind of genome sequences of *Bacteria* or *Archaea*.

This also holds for genome sequences, which do not contain a 16S rRNA gene sequence and whose closest neighbors thus cannot automatically be detected by the TYGS. The reliance on 16S rRNA gene sequences may be regarded as a limitation of the TYGS but, in addition to the computational efficiency of the approach, the use of the 16S rRNA gene cannot currently be dispensed with because the TYGS mainly addresses the needs of taxonomists who intend to describe new species. Despite the effort of large-scale genome sequencing projects such as GEBA and its follow-up projects[15] still ~ 50% of all type strains lack a genome sequence (as of November 2018). In order to avoid creating later heterotypic synonyms[26], type strains that lack a genome sequence, but are at least as closely related to some

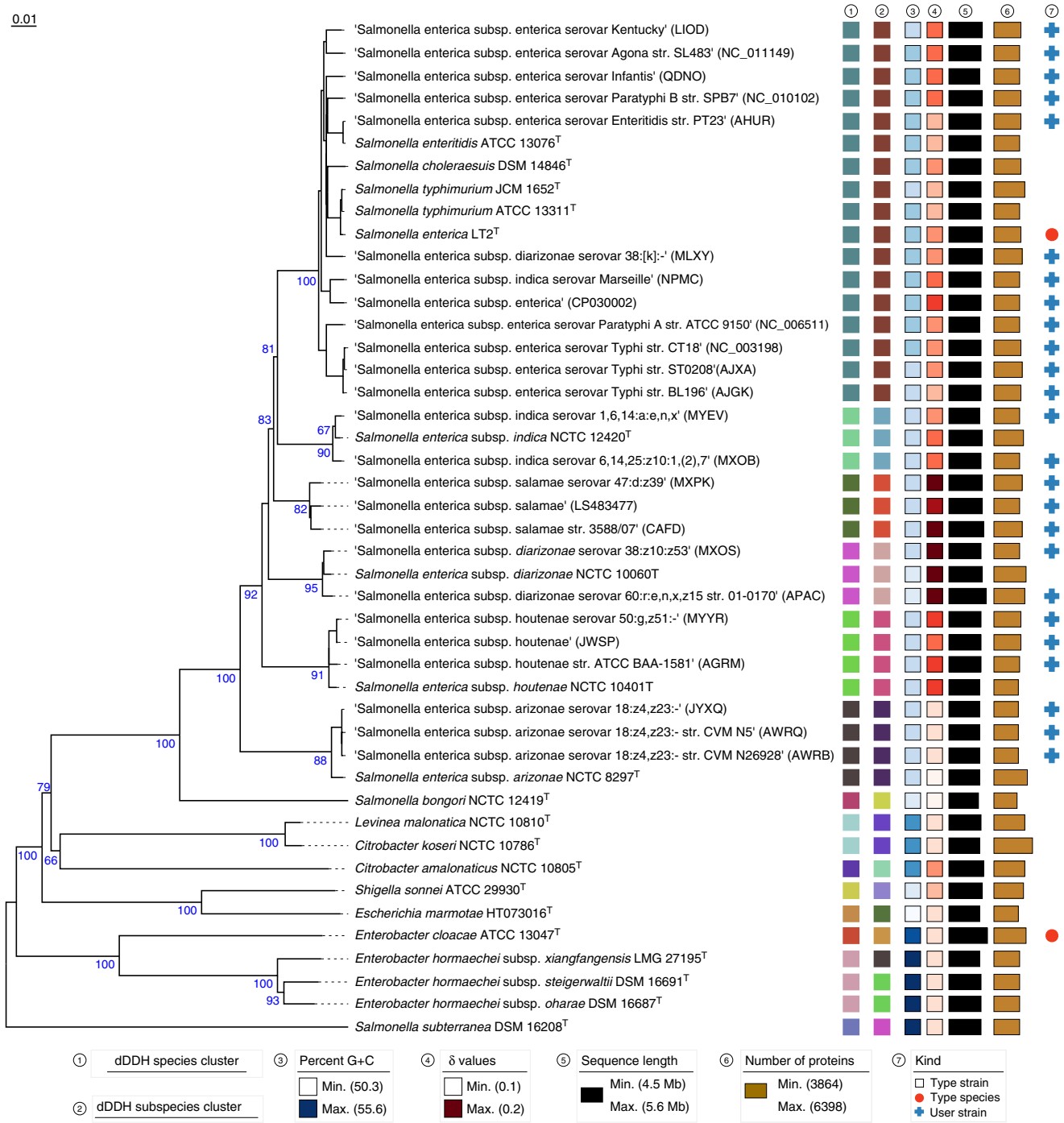

**Fig. 3** TYGS result for the *Salmonella* data set. Tree inferred with FastME 2.1.4[56] from GBDP distances calculated from genome sequences. Branch lengths are scaled in terms of GBDP distance formula $d_5$; numbers above branches are GBDP pseudo-bootstrap support values from 100 replications. Leaf labels are annotated by affiliation to species (1) and subspecies (2) clusters, genomic G+C content (3), $\delta$ values (4), overall genome sequence length (5), number of proteins (6), and the kind of strain (7). User-provided GenBank accession IDs are shown in parentheses; master record accessions are truncated

user-defined query genome sequence as the closest type strains represented by a genome sequence, must be detected by the service and reported to the user. This holds for additional type strains that may affect the creation of species and subspecies as well as those that may affect taxa of higher rank only. In contrast to genome sequences, the 16S rRNA gene is almost comprehensively sampled for type strains of species or subspecies with validly published names, and the TYGS runs queries not only against genome-derived 16S rRNA gene sequences but against the entire collection. For this reason, even phylum-wide genome-

based reclassification approaches still have to make use of 16S rRNA gene sequences[5,25].

Taxon descriptions based on the rules of nomenclature[26], particularly those using cultivated strains[28], remain to be of uttermost importance in the postgenomic era and represent a rapidly expanding field of research in its own right[29]. Among other reasons, such as the importance of experiments with cultivated organisms for detecting gene function[48–50], decent taxon names provide the backbone for a taxonomic classification that can be used by other researchers to bin their metagenomic data[15].

This also holds for the availability of full-length 16S rRNA gene sequences, as metagenomic studies interested in organismic diversity often still only use 16S rRNA gene amplification for greater efficiency[51]. Even nowadays many taxonomic descriptions of new species often start with obtaining a full-length 16S rRNA gene sequence from a strain of interest to detect how likely the strain will yield a new species and whether genome sequencing is accordingly worth pursuing[38]. Thus, it is reasonable to expect a genome sequence used for the description of a new species to be accompanied by a decent 16S rRNA gene sequence; if otherwise, it should better not be used for this purpose. We do not expect this to change if the International Code of Nomenclature of Prokaryotes[26] is modified in the future to accept genome sequences as type material instead of cultivated strains[28] because genome sequences used for this purpose must be of particularly high quality. Continual improvements in sequencing technologies[52] point into the same direction.

Whereas the TYGS offers pairwise similarity calculation[38] and a standard phylogenetic approach including multiple sequence alignment and analysis under the maximum-likelihood and maximum parsimony criteria[10] for the enlarged 16S rRNA gene data sets, which can include closely related type strains not yet represented by genome sequences, the TYGS analysis of genome sequences requires a more-careful method choice for reasons of computational efficiency. A scalable service for genome-based taxonomy needs to process user requests in time. GBDP, the Genome BLAST Distance Phylogeny approach[7,11,13,53], was chosen because it can be used to rapidly infer trees with branch support values from pairwise whole-genome or single-gene distances, which at the same time serve for calculating dDDH values. These dDDH values are in turn used for species and subspecies delineation as previously introduced in the popular Genome-to-Genome Distance Calculator (https://ggdc.dsmz.de) web server 2.1[7,54]. Pairwise distances between database type strain genomes are calculated independently of, and prior to, user requests, thus further decreasing their processing time[14].

Distance-based methods for phylogenetic inference are statistically consistent and yield a huge speed advantage over maximum-likelihood methods[55,56]. Particularly, the FastME approach, which is employed by the TYGS, shows excellent topological accuracy in benchmark studies, outperforming neighbor joining and other distance methods[55,57]. Distance matrices can also be used to calculate statistics such as $\delta$ values[13,33], which allow for assessing the impact of individual operational taxonomic units (OTUs) on overall treelikeness. OTUs with high $\delta$ values render the data less treelike, which can be owing to sequence incompleteness or sequence contamination or be related to long-branch attraction, a typical artifact in phylogenetic inference. Thus, $\delta$ values provide guidance for users regarding the suitability of specific query genome sequences and the reliability of the phylogenetic outcome.

A maximum-likelihood analysis of genome-scale data not only required considerably more time for phylogenetic inference itself—including bootstrapping—but also for the previous detection of clusters of orthologous genes and multiple sequence alignment for all of these clusters[24]. Such computational pipelines are demanding even when reduced to a preconceived selection of few genes[58], which also depends on the relative suitability of the chosen genes and does not fulfill the promises of a genome-based taxonomy[45]. GBDP replaces this entire computational pipeline by conducting pairwise whole-genome comparisons, which account for paralogous genes[11], low-complexity regions[13], and unbalanced genome sizes[54]. Various studies used GBDP for prokaryote, virus, and eukaryote phylogeny and taxonomy[6,10,15–20,22,23]. Although benchmark studies indicated that GBDP outperformed alternative methods in

reflecting evolutionary relatedness without multiple sequence alignment[21], empirical studies did not reveal conflict between GBDP trees and trees inferred from concatenated multiple sequence alignments[23,24] even in the rare cases of conflict between 16S rRNA gene and GBDP trees[5,25]. These comparisons also indicated that GBDP branch support values are conservative, which may be related to the fact that GBDP greedy-with-trimming pseudo-bootstrapping[14] is algorithmically closer to bootstrapping entire genes in concatenated multiple sequence alignments rather than single alignment positions. It was proposed that such a so-called partition bootstrapping can reduce incongruities between trees and thereby provide more realistic support values from phylogenomic data sets compared with ordinary bootstrapping[5,24,25,59]. As conflicts between traditional and genome-based classification regularly appear to be based on over-interpreting largely unresolved 16S rRNA gene trees[5,25], how to calculate branch support is of considerable relevance for constructing a reliable taxonomic classification.

dDDH[7,54], original ANI[60], OrthoANI[30], JSpecies[61] (ANIb and ANIm), gANI[62], and several ANI derivatives on Github are all methods for genome-based species delineation that represent overall genome relatedness indices (OGRI)[63,64]. Although GBDP is the only method among these OGRI implementations that yields branch support for phylogenetic inference and the only one that has been proposed for bacterial subspecies delineation[10], one wonders whether it is also optimal for species delineation. Yet, in addition to the GBDP-specific features, the derivation of dDDH from GBDP also incorporates techniques that lack from other OGRI methods. In contrast, to the linear models used to justify original ANI and JSpecies, which do not really fit the conventional DDH data, dDDH was based on a non-linear model applied to a larger empirical data set[7]. This model yields point estimates on the same scale as conventional DDH values together with confidence intervals[7]. It comes as no surprise that dDDH outperformed original ANI and JSpecies regarding the correlation to conventional DDH[7,54]. This correlation, however, is not only the criterion proposed by the ad hoc committee for the re-evaluation of the species definition in bacteriology[8,9] for judging any in silico genome sequence-based bacterial species delineation method but also the criterion used to establish ANI in the first place[60,61]. In violation of the recommendations of the committee, later ANI versions[30,62] were not even examined regarding their correlation with conventional DDH but at most examined regarding their correlation to each other, which increases the degree of indirection. Such an omission may be due to a misinterpretation of the well-known fact that conventional DDH is error-prone, but mimicking results obtained with conventional DDH on average does not imply mimicking any of its pitfalls[6]. It is thus reasonable to conclude that dDDH is the TYGS method of choice for prokaryotic species delineation, using the recommended GBDP settings of the previously established GGDC web tool 2.1[7].

Compared with previous approaches, TYGS thus particularly facilitates the classification and identification of species and subspecies of *Bacteria* and *Archaea* and particularly the valid publication of new names. We believe this to be a unique approach for integrating genomic and taxonomic data that is tremendously useful for microbial systematics.

## Methods

**TYGS main structure**. TYGS consists of three main components: a comprehensive type-related database including genomic, taxonomic and nomenclatural information; a versatile workflow engine that, e.g., processes stored user requests and queries for novel type strain genomes on a weekly basis; and an interface available on the internet (https://tygs.dsmz.de) that makes TYGS freely and publicly accessible to scientists worldwide.

**The TYGS database**. The TYGS database is the central hub for data that are either produced or consumed by the TYGS workflow engine (Fig. 1). The database stores genome sequences and already calculated G+C content values and intergenomic comparisons including dDDH values for all genome-sequenced type strains. The type strains are also linked to information on taxonomy and nomenclature including literature source, priority, synonyms, and alternative strain deposits.

To this end, TYGS uses the taxonomy databases that underlies the PNU service, a compilation of all names of *Bacteria* and *Archaea,* which have been validly published according to the International Code of Nomenclature of Prokaryotes[26] since 1980, a service offered by the Leibniz Institute DSMZ, the German Collection of Microorganisms and Cell Cultures.

**Routine updates of the TYGS database**. Besides workflows triggered by the submission of new user requests, the engine runs tasks on a regular basis (Fig. 1). The DSMZ PNU service updates its lists of validly published taxon names including synonyms, of type strains, of 16S rRNA gene sequences and of literature sources each month following new issues of the International Journal of Systematic and Evolutionary Microbiology. Genome sequences of type strains are searched for in GenBank each week. Quality checks include matching the known 16S rRNA gene sequence of a type strain of the same species or subspecies and rejecting assemblies comprising > 500 contigs[5,25].

**The TYGS workflow engine**. The TYGS workflow engine (Fig. 1) consists of several independent software modules called "services" that are dependent on the TYGS database and its content. These services cover a wide range of tasks such as data processing and different types of analyses (e.g., fetching new user requests from the web server or reconstructing phylogenies based on intergenomic distances). The interplay of all major components is depicted in Fig. 1 and is explained in detail below.

**User request checker**. This service runs in short intervals and checks for new user requests on the web server. If new requests are available, the relevant data are downloaded and stored into the database. If GenBank accessions were provided, this service will attempt to download the data from the NCBI servers. In case of an error (e.g., wrong accessions) this service will inform the user about the problem.

**rDNA extractor**. The rRNA gene extractor (Fig. 1) extracts available 16S rRNA gene sequences from each user-defined genome using RNAmmer[65] because some of the subsequent TYGS services involve calculations that are based on 16S rRNA gene sequences. Only the 16S rRNA gene sequence with the highest RNAmmer score is considered for further analysis.

If none of the genomes provided by the user contains a 16S rRNA gene sequence, TYGS cannot determine the 10 closest type strain genomes as detailed below. In such a situation the TYGS can still be used, as an analysis restricted to the user-defined genomes can be triggered via the respective option in the submission form. The subsequent processing is then identical to the processing of user requests that did not request the detection of closest neighbors.

After the successful import of a new type strain genome, TYGS attempts to extract the rRNA gene sequences as described above. In the November 2018 version of the database, the average sequence length was 1485 bp, corresponding to near full-length 16S rRNA gene sequences. In the few cases of genome sequences from which a 16S rRNA gene sequence cannot be extracted, it is replaced by a previously published 16S rRNA gene sequence from the same type strain. Only if this fails, another type strain from the same genus is determined and its 16S rRNA gene sequence used as placeholder. Placeholders are not used for phylogenetic inference (see below) but only for determining closest neighbors.

**Pairwise SSU BLASTer**. This service conducts NCBI BLAST+runs[66] with default settings on all possible pairs of 16S rRNA gene sequences and stores the results into the database. This does not only include comparisons between type strains but also comparisons between type strains and user genomes. The genomes yielding the BLAST hits with the 50 highest BLAST bitscores are selected for the calculation of GBDP distances.

In order to not neglect taxonomically relevant type strains that are not currently represented by a genome sequence in the database, the service also conducts a BLAST comparison with the complete set of type strain 16S rRNA gene sequences collected from the literature. Those that lack a genome sequence but are judged as taxonomically relevant are added to the complete list of type strain 16S rRNA gene sequences, which can be phylogenetically explored as described below. Taxonomically relevant are those that yield, per user genome, a bitscore at least as high as the lowest of the bitscores that were obtained when using the genome 16S rRNA gene database for selecting type strain genome sequences.

**SSU distance calculator**. On the basis of the results of the Pairwise SSU BLASTer, the SSU Distance Calculator (Fig. 1) calculates GBDP distances between the 16S rRNA gene sequences that were selected using BLAST. If a query genome sequence contains only a partially complete 16S rRNA gene sequence, the server must still be able to detect the closest neighbors. GBDP can successfully be applied to the

analysis of single genes provided its adaptations for dealing with incomplete sequences are used[53]. In contrast to the BLAST bitscore, which reflects sequence similarity as well as sequence length, some GBDP settings are robust against sequence incompleteness[54]. This holds for formula $d_5$[7], which is used in conjunction with BLAST+and the "coverage" algorithm for obtaining 16S rRNA gene distances. As the type strain database comprises full-length and near full-length 16S rRNA gene sequences and because an individual query 16S rRNA gene sequence has always the same length, a length bias of even the preliminary selection of type strain genomes is expected to be negligible. Using GBDP instead of BLAST for compiling the final list of type strain genomes can thus be understood as an additional precaution against a too narrow strain selection. The final precaution is the analysis of the complete list of type strain 16S rRNA gene sequences as described below.

The lowest 10 16S (SSU) rRNA gene GBDP distances between each user strain and the type strains with genome sequences are used to define its closest relatives. The union of the top-10 lists of type strain genome sequences determined for all user-defined genome sequences yields the set of type strains to be included in the phylogenetic analyses of both 16S rRNA gene sequences and whole-genome sequences—the more diverse user-defined data, the larger this set. SSU rRNA gene GBDP distances are also used for phylogenetic reconstruction as described below (see service "Tree builder"). For this purpose, the "coverage" algorithm resamples individual sites within each high-scoring segment pair (HSP) and was chosen because the pseudo-bootstrapping implementations of the GBDP algorithms "trimming" and "greedy" are based on resampling HSPs and thus underestimate branch support when only few HSPs are present[7].

**Genome distance calculator**. The core feature of TYGS is the calculation of highly accurate intergenomic distances using the GBDP method, including 100 bootstrap replicates each[7]. The calculations are done between user-defined genomes and their closest type strain genomes. To accelerate the processing of user queries all comparisons between type strain genomes are calculated prior to, and independent of, user queries. As all pairwise genome comparisons are computationally independent of each other, they are distributed over an array of DSMZ in-house compute clusters in a highly parallel fashion[14]. The resulting data are subsequently used to infer dDDH estimates[7] and genome-based phylogenetic trees as detailed below.

**dDDH calculator**. The dDDH calculator service predicts digital DNA:DNA hybridization (dDDH) values from intergenomic distances for each user strain and its set of ten most closely related type strains with genome sequences. dDDH values are highly reliable estimators for the relatedness of genomes[7], which have several advantages over the various ANI implementations[6,7,54]. Primarily, dDDH yields better correlations to conventional DDH than ANI, even though this was the major criterion used to justify ANI implementations and thresholds for species delineation, in accordance with the recommendations by the ad hoc committee for the re-evaluation of the species definition in bacteriology[8,9]. Algorithmic features and optimizations of dDDH not present in the various ANI implementations are, for instance, the filtering for paralogous genes or the reporting of confidence intervals for each predicted dDDH value[6,7,54].

**Tree builder**. The tree builder service calculates both genome and 16S rRNA gene GBDP trees including branch support[14]. Phylogenies are inferred using FastME 2.1.4 with a BioNJ starting tree and Subtree Pruning and Regrafting postprocessing[55,56]; the search settings for original and bootstrapped distance matrices are identical.

For assessing the treelikeness of the entire distance matrix and the contribution of individual strains, the tree builder service calculates δ statistics[13,33]. Strains that yield exceptionally high δ values can negatively affect phylogenetic inference. The δ values thus point the user to genome sequences that may better be removed from the data set and which may even be contaminated.

**Species and subspecies clusterer**. This service conducts a clustering using the all-against-all genomic distances and the established thresholds[7,10] to cluster the user-defined and most closely related type strain genome sequences at the species and subspecies level. Contrary to arbitrarily chosen clustering algorithms such as complete linkage, average linkage, or single linkage[67], here an approach is preferred that takes existing type strains as well as the priority of their species or subspecies names into account and otherwise minimizes the number of resulting species, in accordance with the rule to avoid introducing new names wherever possible[26]. To this end, strains are ordered by (i) sorting real type strains increasingly according to priority and placing non-type strains last; (ii) breaking ties by sorting strains decreasingly by the number of links, i.e., distances to other strains below the species threshold; (iii) breaking further ties by sorting strains increasingly by the sum of their distances to all other strains. All strains with links to the first strain on this ordered list of real or potential type strains are then assigned to the first species cluster. This assignment is repeated with the second, third etc. not yet assigned real or potential type strain on the list until all strains are assigned to some species cluster. Subspecies clustering works in the same way but with the subspecies-specific threshold[10].

**User results updater**. The results updater checks for recently inferred trees and dDDH values and if available transfers these data to the web server. An e-mail notification is sent to the user at the same time. As the calculations depend on the workload of the computing infrastructure, it can take some time until the results appear but users can monitor the progress using the web link pointing to the TYGS results.

**The TYGS web server**. The TYGS web server (https://tygs.dsmz.de) is free, publicly available on the Internet as an easy-to-comprehend interface for submitting requests and browsing results. An according menu item allows for the rapid submission of a request including an e-mail address and one to several user-defined genomes. A user account is not required because jobs and their results are protected by a universally unique identifier, a long 128-bit number, which is provided confidentially but could nevertheless easily be shared between collaborators. During the submission phase, the user can choose how the data are analyzed.

For instance, the analysis can be restricted to the submitted genome sequences, thus turning off the detection of closest type strain genomes. Type strain genomes can also manually be selected from the complete list of available type strain genomes.

Upon submission the web server checks the FASTA or GenBank files for valid nucleotide data, checks the e-mail address, and determines whether or not the same set of genomes has already been submitted before by the same user. In case of conflict, an error message is displayed at the top of the submission page.

Once the TYGS job has finished, the user receives a confirmation e-mail with a link to the result page. The result page contains tabular data such as dDDH estimates for user genomes and optionally their closest type strain genomes, as well as annotated whole-genome and 16S rRNA gene phylogenetic trees. For type strains of species or subspecies with validly published names, names of synonyms, alternative strain deposits, and literature sources are displayed, if possible with links to sources such as scientific journals and culture collections, and in any case with the option to download the literature in BibTeX format. The trees can be interactively explored in a tree viewer and customized before export. Annotation with δ values points to individual strains that may be problematic for the analysis.

The complete list of type strain 16S rRNA gene sequences, which is compiled as described above, comprises a larger set of genome-derived 16S rRNA gene sequences as well as 16S rRNA gene sequences from type strains at least as closely related to the user-defined strains but as yet lacking a genome sequence. The user can trigger pairwise similarity calculation[38], multiple sequence alignment and phylogenetic analysis of this enlarged data set under the maximum-likelihood and maximum parsimony criteria[10]. The analysis will be run by the DSMZ gene phylogeny pipeline (https://ggdc.dsmz.de/phylogeny-service.php) and its results sent to the chosen e-mail address. This allows for analyzing a broader sampling of the 16S rRNA gene with a standard phylogenetic approach and for recognizing additional type strains, if any, that would need to be taxonomically compared with the user-defined strains.

**Example data sets**. A data set of 24 *Mycobacterium* genomes was collected from the according "complete genomes" page at NCBI (https://www.ncbi.nlm.nih.gov/genomes/GenomesGroup.cgi?taxid = 1763; Accessed 11 November 2018). A second data set of 23 *Salmonella* genomes was collected from the NCBI server in the same way and reduced to at most 10 genomes per species or subspecies (https://www.ncbi.nlm.nih.gov/genomes/GenomesGroup.cgi?taxid = 590; Accessed 11 November 2018). All GenBank accession IDs are specified in the section Data Availability.

**Reporting summary**. Further information on research design is available in the Nature Research Reporting Summary linked to this article.

## Data availability

The exemplary sequence data of the *Mycobacterium* data set analyzed in this study are publicly available in the GenBank repository under the accession IDs MMLR00000000, FPVV00000000, JKER00000000, BDOP00000000, AGAL00000000, BDOB00000000, FOOB00000000, LQIX00000000, CP016640, LJHL00000000, CP012885, MNAM00000000, NKRE00000000, PQOU00000000, LWCL00000000, LZIO00000000, NC_016946, PSQD00000000, LZSX00000000, MBEO00000000, LZLH00000000, HG917972, CP025779, and PQBM00000000.

The exemplary sequence data of the *Salmonella* data set analyzed in this study are publicly available in the GenBank repository under the accession IDs QDNO00000000, CP030002, LIOD00000000, AHUR00000000, AWRB00000000, JYXQ00000000, AWRQ00000000, APAC00000000, MLXY00000000, MXOS00000000, JWSP00000000, MYYR00000000, AGRM00000000, MXOB00000000, NPMC00000000, MYEV00000000, MXPK00000000, LS483477, CAFD00000000, NC_006511, NC_010102, AJGK00000000 and NC_003198. All other relevant data are available upon request.

## Code availability

TYGS is a web service that fulfills end-user requirement of the journal for web tools by providing the service free and publicly accessible with any modern web browser. The underlying TYGS database is implemented in the object-relational database system PostgreSQL (https://www.postgresql.org/). The TYGS workflow engine is written in the programming language Ruby (https://www.ruby-lang.org/en/), whereas the TYGS web server is implemented in Rails 5, a modern web application development framework (https://rubyonrails.org/). The web application makes use of Bootstrap 3.3.7, a free and open-source front-end framework (library) for designing websites and web applications (https://getbootstrap.com/), and PhyD3[68] for the visualization of phylogenetic trees.

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

## Acknowledgements

We kindly thank Manfred Rohde (Helmholtz Centre for Infection Research, Germany) for providing electron micrographs used as illustrations on the TYGS web page. This work was supported by Deutsche Forschungsgemeinschaft within "Sonderforschungs-bereich TRR 51".

## Author contributions

J.P.M.K. and M.G. devised TYGS. J.P.M.K. implemented the TYGS web server. The TYGS workflow engine was implemented by J.P.M.K., except routines for the periodic database updates, which were provided by M.G. J.P.M.K. drafted the manuscript. Both authors finalized the manuscript.

## Additional information

**Competing interests:** The authors declare no competing interests.

