## [Peer Review File · Nature Communications]

Reviewers' comments:

Reviewer #1 (Remarks to the Author):

Meier-Kolthoff and Göker present TYGS, a new web service for species and subspecies identification on the basis of type strains. The service is only available online and utilizes the multiple resources developed at DSMZ. In general, such a tool is useful for microbiologists and could be widely used if shown to be accurate and easy to use. The main contribution of this specific manuscript seems to be the backend implementation of a workflow of existing tools and a web-based front-end that enables users to submit their data and view the results.

The manuscript itself is well written but lacks any information more than a description of the workflow and some (unsubstantiated) claims about the underlying tools. The authors refer to five of their own publications to back up some of these claims, but at least in my opinion, a novel manuscript should also contain some novel results.

This leaves the web server as the pivotal part of this publication and at the for me (I tried using two web browsers) I could not even get the example to work.

If the authors intend to have an impactful publication, I would suggest that they explain their workflow in more details and benchmark it.

For the SSU rRNA part of their workflow, I would propose to benchmark the recovery rates of SSU rRNA genes from incomplete genomes with SSU rRNA genes of differing completeness, the performance of their SSU rRNA distance calculation in comparison to some ground truth (such as full length alignments), and the accuracy of their SSU rRNA tree building routine (which seems distance based) in comparison to a state-of-the-art phylogenetic tree reconstruction method.

For the whole genome comparison part of their workflow, which seems to be at the core of their tool, the authors have to provide convincing evidence that their tool is at least of comparable accuracy to other tools. Though the authors have provided references, I could not find a benchmark of the combined tools.

In general, the manuscript could also benefit from a comparison of different methods for species delineation as proposed by the "ad hoc committee for the re-evaluation of the species definition in bacteriology", including those using protein-coding genes. The authors have lightly discussed this topic, but their discussion seems rather shallow in the current manuscript.

Reviewer #2 (Remarks to the Author):

I thank the authors for providing such a well-constructed web service. Exploration of the BCG example does an excellent job of highlighting the many features of the TYGS and the impressive amount of linked information provided for type material. I do not agree with many of the selected approaches used by the TYGS, but am certain many will find the resource useful in their research.

1. It appears the "Prokaryotic Nomenclature Up-to-date" has not been updated since October, 2017. Is this going to be problematic for continued updating of the TYGS? Is there a viable alternative for obtaining type material information?

2. My attempts to submit a job to the TYGS failed with the message: "The change you wanted was rejected. Maybe you tried to change something you didn't have access to. If you are the application owner check the logs for more information."

3. The genome-based phylogeny provided on the TYGS is inferred under the minimum evolutionary (ME) criteria applied to GBDP distances. It is more common to infer phylogenies using a maximum-

likelihood (ML) model applied to a multiple sequence alignment. Can the authors report on any literature that show general agreement of inferred species relationships between the ME and ML approaches? I believe some discussion of the use of ME over ML in the manuscript would help orientate reads more familiar with the latter.

4. Metagenome-assembled genomes (MAGs) often lack 16S rRNA sequences due to challenges in assembling this gene. The TYGS currently requires a high-quality 16S rRNA gene for it to process a genome. Would it be possible to relax this requirement by either using another marker gene (e.g., a single copy ribosomal protein) or using a fast whole-genome method (e.g., FastANI) to identify close neighbors?

5. Are there any plans to incorporate “candidatus” species? I believe some users may find it confusing to submit a genome, have it reported as a novel species, but later to find out it is similar to a published genome from a candidate species.

Reviewer #3 (Remarks to the Author):

The manuscript by Meier-Kolthoff and Göker describes a webserver, called TYGS, serving microbial systematists to automatically compare genome sequences of new organisms to a growing database (updating weekly) of well-curated type strain genomes. To save on computation time, TYGS initially uses 16S rRNA gene sequences to identify the most closely related type strains and then compares the genomes. The output of TYGS is a matrix of digital DNA/DNA hybridization values, a 16S rRNA gene tree, a Genome-based Distance Phylogeny method (GBDP) tree, some summary statistics, and links to order the related type strains in culture collections. TYGS uses the data to determine whether the query genome is a potentially novel species or subspecies.

The manuscript makes a strong case for the value of TYGS. This tool serves a research community that is variably equipped for large-scale genomic comparisons, yet it is increasingly clear that microbial taxonomy must leverage genomes. The fact that TYGS taps a database focused on type strain genomes that is updated weekly is a great plus – without a database focused on type strains, it can be difficult to get through the “noise” of a large number of genomes to get to comparisons that are critical for taxonomy. Thus, the biggest strengths of TYGS are: (1) comparison to an up-to-date and well-curated database that is focused on microbial taxonomy; (2) easy to use – important to serve a broad international community of taxonomists; (3) clear and useful outputs – including calls on species and subspecies identification and phylogenetic/phylogenomic trees.

Weaknesses

The weaknesses are related to the limited utility of the system. In particular, in my view, this tool mainly serves people interested in species and subspecies-level taxonomy and does not have broad utility. More specifics on the weaknesses are described below.

- Doesn't address higher-level taxonomy well and is therefore of limited value (valuable mainly for microbial taxonomists working on established groups of bacteria and archaea). Most members of the broader microbiology research community (e.g., microbial ecologists) are mostly concerned with higher-level taxonomy. The exclusive focus of TYGS on species and subspecies therefore misses the mark with this broader community. But this comment is also relevant taxonomists working on less crowded parts of the tree of life, which have few type strains to serve as comparisons. TYGS doesn't have anything specific to say about taxonomy above the species level.

- Similarly, TYGS is clearly only focused on genomes from cultivated organisms, again limiting its utility to a broader community. Studies of metagenome-associated genomes (MAGs) is becoming increasingly common and are rapidly outpacing datasets of isolate microbial genomes. Many MAGs have incomplete, incorrect, or no 16S rRNA gene and therefore fail the criteria for TYGS.

- To address the comments above, and to provide alternative, but widely used, taxonomic guides, TYGS would be more useful if it also calculated ANI and AAI.
- I don't completely understand how TYGS treats genomes from non-type strains. It would be useful if it did access genomes from non-type strains – could be toggled on and off to allow extrapolation to larger datasets.

Minor

- I suggest defining the acronym in the title, abstract, early in manuscript

Reviewers' comments

The original reviewer questions/comments are prefixed by “**REVIEWER**” throughout. Our respective responses are given in blue colour after the “**RESPONSE**” mark, which is followed by an indented paragraph named “→ **RESULTING CHANGES**”. Questions (Q) and responses (R) are subsequently indexed by a number to ease cross-referencing throughout this document. References are provided as short PubMed IDs (PMID) whenever available to increase reading flow of this document.

Reviewer #1

REVIEWER (Q1): Meier-Kolthoff and Göker present TYGS, a new web service for species and subspecies identification on the basis of type strains. The service is only available online and utilizes the multiple resources developed at DSMZ. In general, such a tool is useful for microbiologists and could be widely used if shown to be accurate and easy to use. The main contribution of this specific manuscript seems to be the backend implementation of a workflow of existing tools and a web-based front-end that enables users to submit their data and view the results.

RESPONSE (R1): We are grateful for the basically positive evaluation. In our view, even if the TYGS consisted only of the integration of available tools into an easy-to-use platform, the entire system would be more than the sum of its parts. Many highly cited bioinformatics publications did not invent new algorithms but simply made existing approaches available to a broad user community. However, in addition to the integration of existing software quite a few algorithms have been implemented specifically for the TYGS. We hope this is more obvious from the new thoroughly revised version of the manuscript. Details are provided below.

REVIEWER (Q2): The manuscript itself is well written but lacks any information more than a description of the workflow and some (unsubstantiated) claims about the underlying tools. The authors refer to five of their own publications to back up some of these claims, but at least in my opinion, a novel manuscript should also contain some novel results.

RESPONSE (R2): We agree that a platform such as the TYGS is expected to be able to confirm results based on modern genome-based taxonomic approaches but also yield new insights into the classification of microbes. To demonstrate that more clearly, the revised version of the manuscript investigates two datasets containing important pathogens from the genera *Salmonella* and *Mycobacterium*. These exemplars show that the TYGS can reproduce results from recent genome-based taxonomic investigations with ease – as in the case of *Mycobacterium* – and also provide plausible new insights, as in the case of *Salmonella*, for which our analysis demonstrates that the currently recognized subspecies should better be treated as species.

The description of the work flow has been considerably augmented, too. More general questions are addressed in the new and considerably expanded Discussion chapter whereas more technical issues are addressed in the Methods chapter. Details are provided in our responses to more specific concerns raised by the reviewers. While we do

not see a particular problem with backing our claims by our own papers because these were published in peer-reviewed journals (and, wherever we were doing pioneering work, can even be the only possible publications to back up our claims), it should be evident from the other literature sources cited in the revised version of the manuscript that the proposed methods are sound.

→ **RESULTING CHANGES:**

The new Results chapter describes the results from the analyses of the *Salmonella* and *Mycobacterium* data sets. Details are also provided in the other responses.

For the detailed justification for certain tools and methods please see especially the sections of the new Discussion chapter. Details are also provided in the other responses. Features such as treelikeness indicators (δ statistics) and BibTex download of taxonomic literature were added to the TYGS web server in the course of the revision.

Additional explanations were also included in the the Introduction (see revised manuscript with track changes).

REVIEWER (Q3): This leaves the web server as the pivotal part of this publication and at the for me (I tried using two web browsers) I could not even get the example to work.

RESPONSE (R3): We assume that this was a server bug which resulted in the message “*The change you wanted was rejected. Maybe you tried to change something you didn’t have access to. If you are the application owner check the logs for more information.*” It has been fixed in the meantime according to your report (please also see response R10 to reviewer #2). We are sorry for the inconvenience caused and grateful for the bug report.

Such problems are not as easy to spot as local bugs because they depend on the interplay of web browser and server. For this reason, we now checked the TYGS example result pages via the website “browsershots” (<http://browsershots.org>), a page that runs a given URL on more than 100 combinations of different operating systems and browsers (even exotic ones), and the data were always displayed correctly. It is important to run the TYGS web page via the *https* protocol (<https://tygs.dsmz.de>) as indicated in the manuscript and that Javascript is enabled (a warning should show up on the web page if Javascript is disabled). Finally, we have a new test system in place which automatically renders the pages of the TYGS website as images (including the examples) and compares them to expected reference images. In that way, problems on the TYGS web page are automatically detected and reported to us in timely manner.

→ **RESULTING CHANGES:**

A bug affecting the correct display of user results on the TYGS web server has been fixed and scripts for extended testing of the server were implemented in order to rule out such bugs in the future.

REVIEWER (Q4): If the authors intend to have an impactful publication, I would suggest that they explain their workflow in more details and benchmark it.

RESPONSE (R4):

We agree that the work flow and the methods require a more detailed explanation and justification in the manuscript. More details are given below (see responses to Q5-Q7).

→ RESULTING CHANGES TO THE REVISED MANUSCRIPT:

We have updated the Introduction and Methods chapter of the manuscript and added a completely new Discussion chapter (see response to Q5-Q7). The TYGS is also better illustrated with examples in the new Results chapter.

REVIEWER (Q5): For the SSU rRNA part of their workflow, I would propose to benchmark the recovery rates of SSU rRNA genes from incomplete genomes with SSU rRNA genes of differing completeness, the performance of their SSU rRNA distance calculation in comparison to some ground truth (such as full length alignments), and the accuracy of their SSU rRNA tree building routine (which seems distance based) in comparison to a state-of-the-art phylogenetic tree reconstruction method.

RESPONSE (R5):

An appropriate answer appears to be two-fold.

(a) TYGS SSU rRNA gene trees

The suitability of certain settings of the GBDP method for analyses of single genes has been demonstrated in the literature (PMID: 21037964), much like the suitability of GBDP distance formulas d4 and d5 for accurately representing sequence dissimilarity even in the case of far from complete sequences (PMID: 21304684). The GBDP 16S rRNA gene tree displayed by the TYGS has nevertheless not been intended as an ultimate phylogenetic outcome but as a rapid method for assessing the information content of this gene for the strains under study. This works well because the GBDP distances used to infer the phylogeny are also used to compile the list of type-strain genomes to be compared, i.e. the results from the more time-consuming parts are re-used.

The revised version of the manuscript hopefully clarifies that the TYGS contains a switch for additionally inferring maximum-likelihood and maximum-parsimony 16S rRNA gene trees. This feature also allows users to detect type strains of closely related species and subspecies which are not yet represented by a genome sequence – which is particularly important for drawing taxonomic consequences – and to study a wider range of type strains with genome sequences. The results are automatically sent by e-mail, including publication-ready descriptive text. Thus the users have the option to replace the GBDP single-gene analysis by the more widely known multiple alignment approach in conjunction with phylogenetic inference based on ML and MP. Attempting to further elucidate the suitability of single-gene phylogenetic analysis with GBDP thus appears to not be necessary.

(b) Relevance of incompleteness of 16S rRNA gene sequences

The reason why we gave 16S rRNA gene sequences a role in selecting the closest reference genome sequences is that the current version of the TYGS is mainly targeted towards taxonomists who intend to describe new species. As hopefully clear from the revised version of the manuscript, this is an expanding field of research in itself, even in comparison to the current effort directed towards metagenomics. (See also response to Q16.)

Like many others we believe that taxon descriptions based on the rules of nomenclature, particularly those using cultivated strains, remain to be of uttermost importance in the postgenomic era (PMID: 28731846, 22770837, 28507541, 22661685). Among other reasons, such as the importance of experiments with cultivated organisms for detecting gene function, decent taxon names provide the backbone for a taxonomic classification that can be used by other researchers to bin their metagenomic data. This also holds for the availability of full-length 16S rRNA gene sequences, since many metagenomic studies interested in organismic diversity still only use 16S rRNA gene amplification for greater efficiency (PMID: 19201692).

More importantly, about 50% of the type strains still do not have a genome sequence. To rule out the creation of later heterotypic synonyms, the 16S rRNA gene is thus mandatory to detect species or subspecies w/o a genome sequence that are at least as closely related to the user-defined strains than the closest ones with a genome sequence. We hope that the revised manuscript clarifies this issue and also clarified how the TYGS enables the user to detect such instances. The users can request an additional 16S rRNA gene phylogenetic analysis (ML and MP) for the primarily chosen genomic 16S rRNA reference genes as well as the 16S rRNA genes from type strain that as yet lack a genome sequence also stored in the DSMZ nomenclature database. This addition to the TYGS service not only enables the user to detect closely related species that lack a genome sequence but also yields a more comprehensive selection of type strains with genome sequences. These could be manually chosen in a second query if users opined that they were of interest.

Thus there are good reasons to expect a genome sequence supposed to accompany a new species description to contain a decent 16S rRNA gene sequence; if otherwise, it should simply not be used for this purpose. This is not expected to change if the code of nomenclature is modified in the future to accept genome sequences as type material instead of cultivated strains, as some microbiologists have suggested (PMID: 25769508), because in that case the genome sequence must be of particularly high quality. The use of 16S rRNA gene sequences by the current version of TYGS is thus not on oversight of the authors but a deliberate choice. In our experience most taxonomic descriptions of new species even nowadays start with obtaining a full-length 16S rRNA gene sequence, and genome sequencing is conducted only if the 16S rRNA gene analysis indicates that it is worth pursuing the taxonomy of the given strain.

If a query genome sequence contains only a partially complete 16S rRNA gene sequence the server is still expected to detect the closest neighbours. This holds because the final list of reference genome sequences is compiled using a distance function that is robust against sequence incompleteness (PMID: 21304684). While similarity indexes such as the BLAST bitscore are dependent on sequence similarity and sequence length, the bitscore is only used for compiling the preliminary list of reference genome sequences, which is then

examined in further detail. More importantly, the reference database comprises full-length and near full-length 16S rRNA gene sequences. As an individual query 16S rRNA gene sequence has always the same length, a length bias of even the preliminary selection is thus expected to be negligible.

→ **RESULTING CHANGES (Methods):**

Technical issues related to topics (a) are now addressed in the extended Methods chapter. The first paragraphs of the new Discussion explain in detail the continuing need for 16S rRNA gene sequences even in the current area of genome-based taxonomy, thus addressing topic (b).

REVIEWER (Q6): For the whole genome comparison part of their workflow, which seems to be at the core of their tool, the authors have to provide convincing evidence that their tool is at least of comparable accuracy to other tools. Though the authors have provided references, I could not find a benchmark of the combined tools.

RESPONSE (R6):

We have divided our response in three sections.

(a) Performance of the combined tools

Species delineation and phylogenetic inference both rely on GBDP distance calculation and are displayed together for convenience but are otherwise independent of each other.

(b) GBDP for obtaining dDDH values and according benchmark study

It was demonstrated in our previous studies (PMIDs: 23432962, 19201692) that, according to the criterion put forward by the “ad hoc committee for the re-evaluation of the species definition in bacteriology”, dDDH outperforms other methods for species delineation of *Archaea* and *Bacteria*. For more details please see responses to Q7 and Q18.

(b) GBDP genome-scale phylogenies

The GBDP tool was established as genome-scale phylogenetic method quite some time ago (PMIDs: 15166018, 16854218) and was routinely applied to prokaryote taxonomy at the nucleotide and protein level (PMIDs: 28604660, 28066339, 30186281, 27670113, 27375597, 26915094, 26373441, 25780495, 28106881). Benchmark studies already demonstrated the suitability of FastME in phylogenetic inference (PMIDs: 14694080, 26130081), whereas others demonstrated that GBDP distances outperform alternative methods for obtaining evolutionary distances (PMID: 23843191). We do not believe topological accuracy alone to be crucial in phylogenetic inference. Rather, a suitable approach for calculating branch support is needed. Empirical studies indicate that GBDP support values are conservative (PMID: 23432962, DOI: 10.1002/cpe.3112), which is related to the observation that bootstrapping genes instead of individual alignment positions yields more cautionary measures of branch support in phylogenetic analyses of concatenated multiple sequence alignments (PMID: 28106881, DOI: 10.1111/j.1096-0031.2009.00295.x).

A crucial issue for the TYGS is that we need to balance speed and accuracy. We also believe that a full maximum-likelihood (ML) analysis including bootstrapping is preferable from a purely phylogenetic viewpoint. But a full ML analysis not only needs way more time than a FastME analysis but also requires the detection of clusters of orthologous genes and multiple sequence alignment for all of them. Thus the computational effort needed for running the full pipeline needs to be taken into account, not only for the final ML analysis. Moreover, it would not be particularly efficient to generate intergenomic distances for species and subspecies delineation on the one hand and then running an independent phylogenetic analysis on the other hand. The GBDP method has the probably unique advantage of yielding (sub-)species boundaries as well as decent phylogenetic trees with conservative measures of support.

Furthermore, distance methods have certain specific strengths even in a purely phylogenetic context. We added the reporting of delta statistics as introduced by Holland et al. (2002) (PMIDs: 12446797, 16854218) to the TYGS. These allow for the assessment of the treelikeness of distance matrices and of the impact of individual OTUs on this treelikeness. The lower the delta value, the higher the treelikeness. High delta values of certain OTUs (strains, genome sequences, 16S rRNA genes) can be due to sequence incompleteness or sequence contamination or be related to long-branch attraction, a typical artefact in phylogenetic inference. Thus delta values provide guidance for users regarding the suitability of their query genome sequences.

→ RESULTING CHANGES TO THE REVISED MANUSCRIPT:

According information for justifying our choice of methods was added to the middle section of the new Discussion chapter. Technical details are provided in the revised Methods chapter. The use of Delta values is illustrated by the new examples and referred to again in the Discussion.

REVIEWER (Q7): In general, the manuscript could also benefit from a comparison of different methods for species delineation as proposed by the “ad hoc committee for the re-evaluation of the species definition in bacteriology”, including those using protein-coding genes. The authors have lightly discussed this topic, but their discussion seems rather shallow in the current manuscript.

RESPONSE (R7):

The 2002 paper by Stackebrandt et al. (DOI: 10.1099/ijs.0.02360-0) emphasized that “*Investigators are encouraged to propose new species based upon other genomic methods or techniques provided that they can demonstrate that, within the taxa studied, there is a sufficient degree of congruence between the technique used and DNA–DNA reassociation.*” To date digital DDH (dDDH), which is the method provided since several years by the highly cited GGDC web service and now available via the TYGS, is the method that showed the highest congruence with traditional DNA-DNA hybridization/reassociation. We are not aware of any methods based specifically on protein-coding genes that have even been tested in this respect, let alone were shown to outperform dDDH. In fact, we do not read Stackebrandt et al. (2002) as preferring methods based on single genes or multiple genes over methods based on genome sequences. For this reason, our approaches are in full accordance with the recommendations by the ad

hoc committee, as already emphasized in our previous publications. We believe that the relevant comparisons of different methods for species delineation were already conducted (e.g., PMID: 23432962, 19201692).

→ **RESULTING CHANGES TO THE REVISED MANUSCRIPT:**

For greater clarity the report of the ad hoc committee is now cited in the Introduction and again referred to in the last paragraphs of the entirely new Discussion chapter.

Reviewer #2 (Remarks to the Author)

REVIEWER (Q8): I thank the authors for providing such a well-constructed web service. Exploration of the BCG example does an excellent job of highlighting the many features of the TYGS and the impressive amount of linked information provided for type material. I do not agree with many of the selected approaches used by the TYGS, but am certain many will find the resource useful in their research.

RESPONSE (R8): We are grateful for such a positive evaluation. We think, however, that the computational methods used by the server provide an excellence balance between speed and accuracy and are particularly optimal for bacterial species delineation. We hope that the revised version of the manuscript clarifies these methodological issues. For instance, the analysis of two new datasets (*Salmonella* and *Mycobacterium*) are treated in the new Results chapter of the manuscript to illustrate various aspects of using the TYGS.

REVIEWER (Q9): It appears the “Prokaryotic Nomenclature Up-to-date” has not been updated since October, 2017. Is this going to be problematic for continued updating of the TYGS? Is there a viable alternative for obtaining type material information?

RESPONSE (R9): We fully agree with this concern. Yet the data on nomenclature used by TYGS are lagging behind the last IJSEM issue for at most one week. The internal databases of DSMZ and those used by the TYGS are not dependent on the status of the DSMZ “Prokaryotic Nomenclature Up-to-date” (PNU) website. In fact, it is the other way around. Because the online version of PNU is currently lagging behind for months, the TYGS team took over data collection in 2018 in order to speed-up data delivery. Right now other departments are taking care of the according adaptations of the DSMZ-internal data flow and the data delivery on the DSMZ nomenclature website. (These departments are not to blame for the delay either because a lot of modifications are necessary to satisfy everybody involved, including the curators of the strain collection.) The major internal release has been deployed last week and we suppose that the PNU website is soon up-to-date again. However, TYGS is not even affected right now and already up-to-date. Please note that this is DSMZ-internal information which would not like to disclose in our manuscript; we just added a general explanation to the discussion.

→ RESULTING CHANGES:

We added to the Methods chapter that The TYGS type-strain genome database is updated automatically on a weekly basis, whereas the nomenclature database is updated each month.

REVIEWER (Q10): My attempts to submit a job to the TYGS failed with the message: “The change you wanted was rejected. Maybe you tried to change something you didn’t have access to. If you are the application owner check the logs for more information.”

RESPONSE (R10): This was a server bug that has been fixed in the meantime according to your report (please also see response R3 to reviewer #1). We are sorry for the inconvenience caused and grateful for the bug report.

Such problems are not as easy to spot as local bugs because they depend on the interplay of web browser and server. For this reason, we now checked the TYGS example result pages via the website “browsershots” (<http://browsershots.org>), a page that runs a given URL on more than 100 combinations of different operating systems and browsers (even exotic ones), and the data were always displayed correctly. It is important to run the TYGS web page via the *https* protocol (<https://tygs.dsmz.de>) as indicated in the manuscript and that Javascript is enabled (a warning should show up on the web page if Javascript is disabled). Finally, we have a new test system in place which automatically renders the pages of the TYGS website as images (including the examples) and compares them to expected reference images. In that way problems on the TYGS web page are automatically detected and reported to us in timely manner.

→ **RESULTING CHANGES:**

A bug affecting the correct display of user results on the TYGS web server has been fixed and scripts for extended testing of the server were implemented in order to rule out such bugs in the future.

REVIEWER (Q11): The genome-based phylogeny provided on the TYGS is inferred under the minimum evolutionary (ME) criteria applied to GBDP distances. It is more common to infer phylogenies using a maximum-likelihood (ML) model applied to a multiple sequence alignment. Can the authors report on any literature that show general agreement of inferred species relationships between the ME and ML approaches? I believe some discussion of the use of ME over ML in the manuscript would help orientate reads more familiar with the latter.

RESPONSE (R11):

This is indeed a valuable suggestion. We have addressed this topic in more detail in the revised version of the manuscript. The main issue here is that we need to balance speed and accuracy. We agree that a full ML analysis including bootstrapping is preferable from a purely phylogenetic viewpoint. But a full ML analysis not only needs way more time than a FastME analysis but also requires the detection of clusters of orthologous genes and multiple sequence alignment for all of them. Thus the computational effort needed for running the full pipeline needs to be taken into account, not only for the final ML analysis. Moreover, it would not be particularly efficient to generate intergenomic distances for species and subspecies delineation on the one hand and then running an independent phylogenetic analysis on the other hand. The GBDP method has the probably unique advantage of yielding (sub-)species boundaries as well as decent phylogenetic trees with conservative measures of support.

The GBDP tool was established as genome-scale phylogenetic method quite some time ago (PMIDs: 15166018, 16854218) and was routinely applied to prokaryote taxonomy at the nucleotide and protein level (PMIDs: 28604660, 28066339, 30186281, 27670113, 27375597, 26915094, 26373441, 25780495, 28106881). Benchmark studies already demonstrated the suitability of FastME in phylogenetic inference, whereas others demonstrated that GBDP distances outperform alternative methods for obtaining evolutionary distances (PMID: 23843191). We do not believe topological accuracy alone to be crucial in phylogenetic inference. Rather, a suitable approach for calculating branch

support is needed. Empirical studies indicate that GBDP support values are conservative (PMID: 23432962, DOI: 10.1002/cpe.3112), which is related to the observation that bootstrapping genes instead of individual alignment positions yields more cautionary measures of branch support in phylogenetic analyses of concatenated multiple sequence alignments (PMID: 28106881, DOI: 10.1111/j.1096-0031.2009.00295.x).

Furthermore, distance methods have certain specific strengths even in a purely phylogenetic context. We added the reporting of delta statistics as introduced by Holland et al. (2002) (PMIDs: 12446797, 16854218) to the TYGS. These allow for the assessment of the treelikeness of distance matrices and of the impact of individual OTUs on this treelikeness. The lower the delta value, the higher the treelikeness. High delta values of certain OTUs (strains, genome sequences, 16S rRNA genes) can be due to sequence incompleteness or sequence contamination or be related to long-branch attraction, a typical artefact in phylogenetic inference. Thus delta values provide guidance for users regarding the suitability of their query genome sequences.

→ RESULTING CHANGES TO THE REVISED MANUSCRIPT:

The middle section of the entirely new Discussion chapter is devoted to our choice of methods for phylogenetic inference from genome-scale data. The use of Delta values is illustrated by the new examples and referred to again in the Discussion.

REVIEWER (Q12): Metagenome-assembled genomes (MAGs) often lack 16S rRNA sequences due to challenges in assembling this gene. The TYGS currently requires a high-quality 16S rRNA gene for it to process a genome. Would it be possible to relax this requirement by either using another marker gene (e.g., a single copy ribosomal protein) or using a fast whole-genome method (e.g., FastANI) to identify close neighbours?

RESPONSE (R12):

An appropriate answer appears to be two-fold.

(a) Distinct ways to use the TYGS

In the TYGS environment the presence of a 16S rRNA gene is currently only needed for the automated detection of the closest neighbours of a user genome. Users who have already compiled their own set of genomes can always directly make use of the server. Users who only know the species they want to analyse can select the according type-strain genomes via the TYGS user interface. Thus even if 16S rRNA genes are completely absent from a user genome sequence the current version of the TYGS can still be used, it is just slightly more demanding for the user.

Moreover, if a query genome sequence contains only a partially complete 16S rRNA gene sequence the server is still expected to detect the closest neighbours. This holds because the final list of reference genome sequences is compiled using a distance function that is robust against sequence incompleteness. While similarity indexes such as the BLAST bitscore are dependent on sequence similarity and sequence length, the bitscore is only used for compiling the preliminary list of reference genome sequences, which is then examined in further detail. More importantly, the reference database comprises full-length

and near full-length 16S rRNA gene sequences. As an individual query 16S rRNA gene sequence has always the same length, a length bias of even the preliminary selection is thus expected to be negligible.

(b) Continuing importance of cultivated strains and 16s rRNA gene sequences

Future versions of the TYGS are intended to include alternative methods for detecting the closest genome sequence. The reason why we did not include them right now is that the current version of the TYGS is targeted mainly towards taxonomists who intend to describe new species. As hopefully clear from the revised version of the manuscript, this is an expanding field of research in itself, even in comparison to the current effort directed towards metagenomics. The number of validly published species names has steadily increased in the last decades and now reaches more than 900 per year. (See also response to Q16.) Additionally, the International Journal of Systematic and Evolutionary Microbiology, in which most descriptions of new species are published, recently made it mandatory for such descriptions to be accompanied by a genome sequence.

Like many others we believe that taxon descriptions based on the rules of nomenclature, particularly those using cultivated strains, remain to be of uttermost importance in the postgenomic era (PMID: 28731846, 22770837, 28507541, 22661685). Among other reasons, such as the importance of experiments with cultivated organisms for detecting gene function, decent taxon names provide the backbone for a taxonomic classification that can be used by other researchers to bin their metagenomic data. This also holds for the availability of full-length 16S rRNA gene sequences, since many metagenomic studies interested in organismic diversity still only use 16S rRNA gene amplification for greater efficiency.

Crucially, about 50% of the type strains still do not have a genome sequence. To rule out the creation of later heterotypic synonyms, the 16S rRNA gene is thus mandatory to detect species or subspecies w/o a genome sequence that are at least as closely related to the user-defined strains than the closest ones with a genome sequence. We hope that the revised manuscript clarifies this issue and also shows how the TYGS enables the user to detect such instances. The users can request an additional 16S rRNA gene phylogenetic analysis (ML and MP) for the primarily chosen genomic 16S rRNA reference genes as well as the 16S rRNA genes from type strain that as yet lack a genome sequence also stored in the DSMZ nomenclature database. This addition to the TYGS service not only enables the user to detect closely related species that lack a genome sequence but also yields a more comprehensive selection of type strains with genome sequences. These could be manually chosen in a second query if users opined that they were of interest.

Thus there are good reasons to expect a genome sequence supposed to accompany a new species description to contain a decent 16S rRNA gene sequence; if otherwise, it should simply not be used for this purpose. This is not expected to change if the code of nomenclature is modified in the future to accept genome sequences as type material instead of cultivated strains, as some microbiologists have suggested (PMID: 25769508), because in that case the genome sequence must be of particularly high quality. The use of 16S rRNA gene sequences by the current version of TYGS is thus not on oversight of the authors but a deliberate choice, mainly for reasons of efficiency. In our experience most taxonomic descriptions of new species even nowadays start with obtaining a full-length

16S rRNA gene sequence, and genome sequencing is conducted only if the 16S rRNA gene analysis indicates it is worth pursuing the taxonomy of the given strain.

→ **RESULTING CHANGES TO THE REVISED MANUSCRIPT:**

Statistics on the number of new species per year and further comments have been added to the revised Introduction to emphasize the demand for a platform such as the TYGS. The first paragraphs of the new Discussion explain in detail the continuing need for 16S rRNA gene sequences even in the current area of genome-based taxonomy. The purpose and use of the ML/MP analyses of the expanded 16S rRNA gene data sets are illustrated by the new examples and again referred to in the new Discussion. More details have been added to the Methods chapter.

REVIEWER (Q13): Are there any plans to incorporate “candidatus” species? I believe some users may find it confusing to submit a genome, have it reported as a novel species, but later to find out it is similar to a published genome from a candidate species.

RESPONSE (R13):

The focus of the current version of TYGS on type strains of species and subspecies with validly published names ensures that strains with standing in nomenclature are considered (DOI: 10.1099/ijsem.0.000778) together with their priority. (The revised version of the TYGS makes the taxonomic literature directly accessible in BibTex format, in addition to the previously provided links.) Technically it is no problem to include proposed type strains of species that have no validly published names, and some of them are even contained in the database (they automatically get the lowest priority in the nomenclature-aware clustering). However, by construction the names of such taxa are not centrally collected because the purpose of the central collection (in IJSEM) is precisely to get them validly published.

The category *Candidatus* is not covered by the Bacteriological Code (PMID: 26596770). This implies that a *Candidatus* name can never obtain priority over a validly published name. For this reason, no taxonomic action would be needed if a validly published species later on turns out to be identical at the species level with a previously proposed *Candidatus* name. It also implies, unfortunately, that there is no list of “accepted” *Candidatus* names.

To illustrate the difficulties in assembling a database of *Candidatus* names, we have screened the IJSEM notification lists for the *Candidatus* category. We found the first occurrence in 1999, a peak of 15 such names in 2004 and a sudden disappearance of the reporting after 2013. Authoritative, nomenclature-centred lists such as those in IJSEM can hardly be replaced by alternatives such as the NCBI taxonomy database for a variety of reasons. For instance, submitting a sequence to NCBI under a certain name is not equivalent to formally proposing that name, even though NCBI appears to accept every kind of name.

The same holds for proposals of taxon names that could be validly published but were never sent in for validation. Oren et al. (2018) (PMID: 29873629) have recently provided counts of such proposals but did not provide the list of names and literature sources.

That said, we would emphasize that users can always upload genome sequences of their own choice, including those from *Candidatus* species. The TYGS will then conduct a whole genome-based phylogenetic analysis in much the same way as for type-strain genomes.

→ **RESULTING CHANGES TO THE REVISED MANUSCRIPT:**

We have clarified this in one of the first paragraphs in the new Discussion chapter of the revised manuscript. Features such as BibTex download of taxonomic literature were added to the TYGS software in the course of the revision, further demonstrating the advantages of a comprehensive database solution for linking genome sequences to taxonomic literature.

Reviewer #3

REVIEWER (Q14): The manuscript by Meier-Kolthoff and Göker describes a webserver, called TYGS, serving microbial systematists to automatically compare genome sequences of new organisms to a growing database (updating weekly) of well-curated type strain genomes. To save on computation time, TYGS initially uses 16S rRNA gene sequences to identify the most closely related type strains and then compares the genomes. The output of TYGS is a matrix of digital DNA/DNA hybridization values, a 16S rRNA gene tree, a Genome-based Distance Phylogeny method (GBDP) tree, some summary statistics, and links to order the related type strains in culture collections. TYGS uses the data to determine whether the query genome is a potentially novel species or subspecies. The manuscript makes a strong case for the value of TYGS. This tool serves a research community that is variably equipped for large-scale genomic comparisons, yet it is increasingly clear that microbial taxonomy must leverage genomes. The fact that TYGS taps a database focused on type strain genomes that is updated weekly is a great plus – without a database focused on type strains, it can be difficult to get through the “noise” of a large number of genomes to get to comparisons that are critical for taxonomy. Thus, the biggest strengths of TYGS are: (1) comparison to an up-to-date and well-curated database that is focused on microbial taxonomy; (2) easy to use – important to serve a broad international community of taxonomists; (3) clear and useful outputs – including calls on species and subspecies identification and phylogenetic/phylogenomic trees.

RESPONSE (R14):

We are grateful for the positive evaluation.

REVIEWER (Q15): The weaknesses are related to the limited utility of the system. In particular, in my view, this tool mainly serves people interested in species and subspecies-level taxonomy and does not have broad utility. More specifics on the weaknesses are described below.

RESPONSE (R15):

These concerns need, of course, to be addressed, and we hope that the revised manuscript does so in a more convincing way. In particular, we need to clarify how large the intended main user base of the current version of TYGS actually is, to what extent this main user base benefits from TYGS, how far other scientists can directly benefit from TYGS right now, and to what extent they can indirectly benefit from the taxonomic work conducted with TYGS. We will address this below by responding to the specific comments.

In general, each individual piece of software is of limited use, even though it may still be a major step forward regarding its specific applicability. We believe the TYGS to be such a major step forward even in its current version, as up to now a system that integrates the features of TYGS is not available for microbial taxonomy.

REVIEWER (Q16): Doesn't address higher-level taxonomy well and is therefore of limited value (valuable mainly for microbial taxonomists working on established groups of bacteria and archaea). Most members of the broader microbiology research community (e.g., microbial ecologists) are mostly concerned with higher-level taxonomy. The exclusive

focus of TYGS on species and subspecies therefore misses the mark with this broader community. But this comment is also relevant taxonomists working on less crowded parts of the tree of life, which have few type strains to serve as comparisons. TYGS doesn't have anything specific to say about taxonomy above the species level.

RESPONSE (R16):

An appropriate answer appears to be two-fold. As for microbial ecologists, please see the response to Q17.

(a) Quantity and quality of descriptions of new taxa and according TYGS user base

We agree that the current version of the TYGS is mainly addressing the needs of microbiologists who describe new species or subspecies. However, the number of validly published species names per year has increased to more than 900 per year in 2018, even if one disregards all new combinations. Moreover, the vast majority of species descriptions are descriptions of species within existing genera. (The revised version of the TYGS makes the taxonomic literature directly accessible in BibTex format, in addition to the previously provided links.) The plots, which are included in the revised supplement, do not indicate a flattening of the curve. Additionally, the International Journal of Systematic and Evolutionary Microbiology, in which most descriptions of new species are published, recently made it mandatory for such descriptions to be accompanied by a genome sequence. We conclude that even if the TYGS served only for species delineation its user base would be considerable.

The three most relevant digital DDH and GGDC studies (PMIDs: 23432962, 21304684 and 21304686) have exceeded 1800 citations according to Google Scholar. The integration of these tools with a database and new functionalities such as phylogenetic trees into a single application, the current version of TYGS, is clearly synergetic and will thus very likely meet an even better approval within the scientific community.

→ RESULTING CHANGES:

Statistics on the number of new species per year and further comments have been added to the Introduction to emphasize the demand for platforms such as the TYGS. Features such as BibTex download of taxonomic literature were added to the TYGS software in the course of the revision, further demonstrating the advantages of a comprehensive database solution for linking genome sequences to taxonomic literature.

(b) Usability of the TYGS for taxon descriptions in less crowded parts of the tree of life

The assignment of species to genera and higher taxa requires a phylogenetic analysis, and we do not think that the TYGS has nothing to offer in this respect. For instance, the revised manuscript includes an analysis of a *Salmonella* data set that demonstrates how the TYGS can be used to recognize whether certain species (*Salmonella subterranea* in that case) do not belong to a certain genus.

Type strains of new species that are phylogenetically isolated from other species will still yield next neighbours in the 16S rRNA gene analysis. The according phylogenomic tree can as well be computed. This tree may just be less well resolved because the signal in nucleotide sequences – even at the genome level – decreases when strains become less closely related. However, the obtained resolution is dependent on the data set, while species and subspecies will always become obvious. Moreover, even if the phylogenomic analysis was unsatisfactory regarding its resolution, the user would still obtain the list of most closely related type-strain genomes and could if necessary analyse them elsewhere.

More importantly, single conserved genes such as the 16S rRNA gene yield more resolution in less crowded parts of the tree of life because in these situations the gene contains more differences between the organisms to be analysed while still not suffering from saturation. For this reason, a 16S rRNA gene tree is likely to be entirely sufficient in such cases. Even more importantly, about 50% of the type strains still do not have a genome sequence. To rule out the creation of genera that later on need to be included in already existing genera because they phylogenetically contain the type species of an already proposed genus analysis of the the 16S rRNA gene is still advisable. To rule out the creation of taxa of higher rank that later on need to be included in already proposed taxa of the same rank because they phylogenetically contain their type genera, analysis of the the 16S rRNA gene is still mandatory. One of the basic principles of the code of nomenclature is to avoid the unnecessary creation of taxon names.

The users can request an additional 16S rRNA gene phylogenetic analysis (ML and MP) for the primarily chosen genomic 16S rRNA reference genes as well as the 16S rRNA genes from type strain that as yet lack a genome sequence also stored in the DSMZ nomenclature database. This feature of the TYGS service not only enables the user to detect closely related species that lack a genome sequence but also yields a more comprehensive selection of type strains with genome sequences. These could be manually chosen in a second query if users opined that they were of interest.

→ RESULTING CHANGES:

The revised Discussion chapter explains that information on the higher classification of established type strains can be directly accessed via the link to the BacDive database. The use of the TYGS in demonstrating, using genome-scale data, that certain species need to be placed in a distinct genus is directly exemplified in the newly analysed *Salmonella* data set in the Results chapter. The continuing need for 16S rRNA gene sequences is explained in the first paragraphs of the new Discussion chapter.

REVIEWER (Q17): Similarly, TYGS is clearly only focused on genomes from cultivated organisms, again limiting its utility to a broader community. Studies of metagenome-associated genomes (MAGs) is becoming increasingly common and are rapidly outpacing datasets of isolate microbial genomes. Many MAGs have incomplete, incorrect, or no 16S rRNA gene and therefore fail the criteria for TYGS.

RESPONSE (R17):

As in the case of Q12 AND Q16, an appropriate answer appears to be two-fold.

(a) Distinct ways to use the TYGS

In the TYGS environment the presence of a 16S rRNA gene is currently only needed for the automated detection of the closest neighbours of a user genome. Users who have already compiled their own set of genomes can always directly make use of the server. Users who only know the species they want to analyse can select the according type-strain genomes via the TYGS user interface. A mixed usage is also possible if only one among several user genome sequences contains a 16S rRNA gene. In that case the user only needs to know that her other query genome sequences are closely related to the more complete one that guides the analysis. Thus even if 16S rRNA genes are completely absent from a user genome sequence the current version of the TYGS can still be used, it is just slightly more demanding for the user.

Moreover, even if a query genome sequence contains only a partially complete 16S rRNA gene sequence the server is still expected to detect the closest neighbours. This holds because the final list of reference genome sequences is compiled using a distance function that is robust against sequence incompleteness. While similarity indexes such as the BLAST bitscore are dependent on sequence similarity and sequence length, the bitscore is only used for compiling the preliminary list of reference genome sequences, which is then examined in further detail. More importantly, the reference database comprises full-length and near full-length 16S rRNA gene sequences. As an individual query 16S rRNA gene sequence has always the same length, a length bias of even the preliminary selection is thus expected to be negligible.

(b) Continuing importance of cultivated strains and 16s rRNA gene sequences

The reason why we gave 16S rRNA gene sequences a role in selecting the closest reference genome sequences is that the current version of the TYGS is mainly targeted towards taxonomists who intend to describe new species. As hopefully clear from the revised version of the manuscript, this is an expanding field of research in itself, even in comparison to the current effort directed towards metagenomics. (See also response to Q16.)

Like many others we believe that taxon descriptions based on the rules of nomenclature, particularly those using cultivated strains, remain to be of uttermost importance in the postgenomic era (PMID: 28731846, 22770837, 28507541, 22661685). Among other reasons, such as the importance of experiments with cultivated organisms for detecting gene function, decent taxon names provide the backbone for a taxonomic classification that can be used by other researchers to bin their metagenomic data. This also holds for the availability of full-length 16S rRNA gene sequences, since many metagenomic studies interested in organismic diversity still only use 16S rRNA gene amplification for greater efficiency (PMIDs: 25284151, 23236140).

Crucially, about 50% of the type strains still do not have a genome sequence. To rule out the creation of later heterotypic synonyms, the 16S rRNA gene is thus mandatory to detect species or subspecies w/o a genome sequence that are at least as closely related to the user-defined strains than the closest ones with a genome sequence. We hope that the revised manuscript clarifies this issue and also clarified how the TYGS enables the user to detect such instances. The users can request an additional 16S rRNA gene phylogenetic

analysis (ML and MP) for the primarily chosen genomic 16S rRNA reference genes as well as the 16S rRNA genes from type strain that as yet lack a genome sequence also stored in the DSMZ nomenclature database. This addition to the TYGS service not only enables the user to detect closely related species that lack a genome sequence but also yields a more comprehensive selection of type strains with genome sequences. These could be manually chosen in a second query if users opined that they were of interest.

Thus there are good reasons to expect a genome sequence supposed to accompany a new species description to contain a decent 16S rRNA gene sequence; if otherwise, it should simply not be used for this purpose. This is not expected to change if the code of nomenclature is modified in the future to accept genome sequences as type material instead of cultivated strains, as some microbiologists have suggested, because in that case the genome sequence must be of particularly high quality. The use of 16S rRNA gene sequences by the current version of TYGS is thus not on oversight of the authors but a deliberate choice. In our experience most taxonomic descriptions of new species even nowadays start with obtaining a full-length 16S rRNA gene sequence, and genome sequencing is conducted only if the 16S rRNA gene analysis indicates that it is worth pursuing the taxonomy of the given strain.

→ RESULTING CHANGES TO THE REVISED MANUSCRIPT:

Statistics on the number of new species per year and further comments have been added to the revised Introduction to emphasize the demand for a platform such as the TYGS. The first paragraphs of the new Discussion explain in detail the continuing need for 16S rRNA gene sequences even in the current area of genome-based taxonomy. The purpose and use of the ML/MP analyses of the expanded 16S rRNA gene data sets are illustrated by the new examples and again referred to in the new Discussion. More details have been added to the Methods chapter.

REVIEWER (Q18): To address the comments above, and to provide alternative, but widely used, taxonomic guides, TYGS would be more useful if it also calculated ANI and AAI.

RESPONSE (R18):

Honestly, we believe the calculation of ANI similarities to add absolutely no value to the TYGS. This may sound harsh but we hope this issue can be clarified by looking into the history of these methods, in particular into their justifications. As obvious from the major ANI publications (Goris et al. 2007, PMID: 17220447; Richter and Roselló-Mora, PMID: 19855009), the sole justification for ANI was its high correlation with traditional DNA:DNA hybridization (DDH), and the sole justification for the (inexact) threshold of c. 95% for species delineation was its rough correspondence with 70% DDH.

The DDH criterion is indeed as recommended in the literature (Wayne et al. 1987, DOI: 10.1099/00207713-37-4-463; Stackebrandt et al. 2002, DOI: 10.1099/ijs.0.02360-0.02360). However, the main dDDH publication (Meier-Kolthoff et al. 2013, PMID: 23432962) has shown that dDDH has a higher correlation with traditional DDH than ANI. Obviously, according to the very criterion that has been used to advocate ANI, dDDH surpasses ANI. The methods have the same purpose and dDDH works better. There are a couple of additional advantages of dDDH, such as being based on the GBDP tool, which

can be used to calculate branch support. While phylogenetic trees could be inferred from ANI similarities after conversion to distances, we have never seen ANI trees with branch-support values.

Additional ANI implementations were introduced in the literature by only comparing them with existing ANI methods, thus increasing the magnitude of indirection. Distinct ANI implementations rarely yield exactly the same results, hence “ANI” appears to be a good brand name but it is not a well-defined methodology. It is also revealing that a correction for paralogs has been introduced only lately to an ANI Method (“OrthoANI”), whereas it was present from the very beginning in the GBDP implementation used to calculate dDDH values in the GGDC web service hosted at DSMZ.

As for AAI, the use of protein sequences can of course yield better resolved phylogenies when strains are less closely related. However, much like ANI the AAI does not appear to be available in an implementation that can be used to calculate branch support values. Thus it would have to be replaced by methods such as proteome GBDP. While proteome GBDP was indeed our major choice for inferring phylogenies in our recent studies (PMID: 30186281, 28066339) the use of AAI in the literature is a different one, namely as pairwise similarities for obtaining genus boundaries. This does not guarantee monophyly of the resulting groups when species become less closely related as it would require the data to be ultrametric (PMID: 24505073). Moreover, while in the case of species delimitation the task was to mimic DDH, the pre-genomic era did not establish a quantitative approach to genus delineation. While one could attempt to estimate a boundary from the existing distribution of genus names, we do not think the literature was as yet able to provide a convincing approach. The method would need to be based on a phylogeny instead of pairwise similarities and for truly maximizing taxonomic conservatism only validly published names would need to be taken into account, as well as their synonyms.

Thus while future versions of the TYGS are intended to provide more genome-based indexes, we have currently chosen to not adopt AAI for taxonomic reasons. Integrating a genus delineation method appears to be beyond the scope of the present study and is intended for a future version. User feedback to be obtained once the server becomes public is supposed to provide additional guidance.

→ **RESULTING CHANGES (Methods):**

The last paragraphs of the new Discussion explain in detail why the methods used by the TYGS are superior to alternatives such as ANI. One of the new examples illustrates how genus boundaries can be explored using the TYGS phylogeny pipeline. More details have been added to the Methods chapter. We omitted discussing AAI though because, as explained above, we do not think it is that well established and because we prefer discussing the features of some software that were deliberately included over discussing features that were deliberately left out.

REVIEWER (Q19): I don't completely understand how TYGS treats genomes from non-type strains. It would be useful if it did access genomes from non-type strains – could be toggled on and off to allow extrapolation to larger datasets.

RESPONSE (R19):

Users can already add any kinds of bacterial genomes to their query. Restrictions only affect the genomes stored in the database, for which pairwise comparisons are computed on a routinely basis, independent of user queries to save time when such queries arrive. Because hardware resources are always a limiting factor, the size of the data sets to be analysed must also be limited by the server. Such limitations can be relaxed in the future but they can never disappear entirely.

The focus of the current version of TYGS on type strains of species and subspecies with validly published names ensures that only strains with standing in nomenclature are considered (PMIDs: 26596770, 17082418), together with their priority. Technically it is no problem to include proposed type strains of species that have no validly published names in the database, and some of them are even already contained (they automatically get marked as such and obtain the lowest priority in the nomenclature-aware clustering). However, obtaining a comprehensive collection of such names is near impossible as, by construction, the names of such taxa are not centrally collected because the purpose of the central collection (in IJSEM) is precisely to get them validly published.

Authoritative, nomenclature-centred lists such as those in IJSEM can hardly be replaced by alternatives such as the NCBI taxonomy database for a variety of reasons. For instance, submitting a sequence to NCBI under a certain name is not equivalent to formally proposing that name, even though NCBI appears to accept every kind of name. The NCBI classification is not authoritative (PMID: 22139910). Among the 50,000+ non-type strain genomes in GenBank a large quantity of genomes is misidentified (PMID: 28005526) and their inclusion into the TYGS database would add noise and could negatively affect the conclusions of any study.

It is even difficult to collect all those proposed taxon names that could be validly published but were never sent in for validation. Oren et al. (2018) (PMID: 29873629) have recently provided counts of such proposals in the literature but did not provide the list of names and literature sources. *Candidatus* taxa are good candidates for including them in the nomenclature database but they ceased to be listed in IJSEM after 2013 (see Q13). Much like Oren et al. (2013) we believe that only slightly more effort would be needed on the side of those researchers who propose taxon names to get them validly published; if so, their genome sequence would rapidly appear in the TYGS database.

As for the nomenclature of organisms that as yet cannot be cultivated, there are two future options. If the code of nomenclature is modified to accept genome sequences as type material instead of cultivated strains, as some microbiologists have suggested (PMID: 25769508), such uncultivated organisms will automatically appear in the TYGS. Of course such a move should only be done if the genome sequence to be used as type material are guaranteed to be of particularly high quality. If the code of nomenclature is not accordingly modified, we must hope for improved cultivation methods (PMID: 28731846, 22770837). Quite some progress was made regarding difficult-to-cultivate bacteria in recent years as, e.g., in the case of *Acidobacteria* (PMID: 28731846).

The TYGS authors cannot decide on these issues. Including those *Candidatus* taxa that can be traced back to a publication and trying to obtain the list of names from Oren et al. (2018) are realistic plans for the next version of TYGS, however.

→ **RESULTING CHANGES TO THE REVISED MANUSCRIPT:**

The continuing need for a taxonomic classification based on the rules of nomenclature is emphasized in the first paragraphs of the new Discussion chapter. These should clarify that a platform such as the TYGS has an important role in future microbial classification independent of the potential changes of the code of nomenclature regarding the need of cultivated strains as nomenclatural types of species and subspecies.

REVIEWER (Q20): I suggest defining the acronym in the title, abstract, early in manuscript

RESPONSE (R20):

Thanks for the valuable suggestion.

→ **RESULTING CHANGES TO THE REVISED MANUSCRIPT:**

We have additionally defined the acronym in the abstract but not in the title because of the journal policy that restricts the length of the title.

Data Availability Statement

TYGS fulfills end-user requirement of the journal for web tools by providing the web server free and publicly accessible with any modern web browser. All exemplary sequence data used in this study is publicly available via the specified GenBank accessions.

Reviewers' comments:

Reviewer #1 (Remarks to the Author):

Meier-Kolthoff and Goeker have resubmitted their manuscript "TYGS: an automated high-throughput platform for state-of-the-art genome-based taxonomy". Even though they claim that their manuscript was revised thoroughly, many points I raised in the first round of reviews have not been addressed. In very general terms I would like to comment on the authors rebuttal:

1) At least to me it seems to be unnecessary and bad style to copy-and-paste responses-to-reviewer-comments. At least I read the comments of other reviewers, both out of interest and in order to gather more ideas to help authors.

2) It is generally accepted that not all reviewer comments are addressed and some can be discussed away within reason. It has been a while that authors have tried to discuss away the vast majority of my comments. As mentioned above, I am trying to help the authors to improve their manuscript but it is also the duty of reviewers to maintain a good quality of scientific publications, so I implore the authors to take reviewer comments a bit more serious in general.

First, I will go through the comments I previously made that have not been addressed properly yet. Afterwards, I will have some comments regarding the revision. I hope this will streamline the review process.

Regarding Q2:

This was not only meant in terms of biological results, but as should be obvious from the phrasing of my comments this also pertains to the quality of the tool. Either the tool is shown to be useful (this includes the examples now listed, but these examples are neither benchmarks against a ground truth nor are they generally applicable) using a dedicated benchmark against competing tools (many of which have been published more recently tools in the TYGS workflow and hence could not possibly be compared before). What the authors have done here was to replace one example by two others. I hope my point is more clear now.

Regarding Q4:

The authors have ignored the main part of this comment, i.e. the term benchmark. No one knows how accurate TYGS is at the current point in time.

Regarding Q5:

If this tool should be useful in a general sense, it should be possible to download a genome from NCBI and run it through TYGS. For this reason I asked the authors to "benchmark the recovery rates of SSU rRNA genes from incomplete genomes". The authors can claim that 16S is an essential gene for genomics but that doesn't make it magically appear within genomes. It is not my job to perform this task, but I have done this a few years ago and I was surprised how many genomes lack a 16S rRNA gene. I.e. I'm completely unsatisfied by the answer of the authors especially as my comments only pertained a quite feasible study that is highly relevant for TYGS.

Regarding Q6:

I don't understand how my comment could be misunderstood but it has not been addressed. TYGS needs to be benchmarked, referring 5+ year old studies is not enough. (Also I don't understand in which context PMIDs: 19201692 is of relevance here). Adding random extra features does not address this comment either.

Regarding Q7:

Again: benchmarking and comparing with other tools for species delineation! I used the search engine

google scholar using the terms "species delineation protein coding marker genes" and successfully found examples and tools for this purpose. They might not be more accurate, but this is for the authors to show and prove.

I hope I could clarify my comments. At the current state I can't tell if this tool is better than the mentioned "GGDC web service". Also I am aware that DSMZ is an authority on the field of microbial taxonomy but as a reviewer I have to use the same standards that I hold everyone to. Hence, I cannot accept any argumentum ab auctoritate here. I hope that the authors take my comments a bit more serious this time and that the next version of their manuscript is more convincing!

New comments:

- a) The two new examples for delineating species need more outside evidence and discussion.
- b) I tried the examples listed on the website and many genomes said something like: "Potential new species detected: your strain 'Bacillus_anthraxis_str_Ames_Ancestor' does not belong to any species found in TYGS database." I guess this is due to the limitation to type strains, but also shows a weakness (that another reviewer mentioned as well): Here it would be very useful to have non-type strains in order to connect with other researchers working on similar bacteria/genomes to decide on a type strain that can be explored in-depth later.

Reviewer #2 (Remarks to the Author):

Thank you for the clear and detailed responses to my previous comments. The format of your rebuttal was also much appreciated. I look forward to making use of the TYGS.

Reviewer #3 (Remarks to the Author):

The authors have responded in detail to my criticisms and, in a few cases, made some improvements to the manuscript. I'm satisfied with their detailed responses, as discussed below. In the end, I think this database will be very important for taxonomists working on new species and subspecies of bacteria, but may not immediately be useful for a wider user community of ecologists. This is still a very significant impact.

General criticism 1: Focus on new species and subspecies rather than higher taxonomic ranks.

The authors respond by stressing the large numbers of new taxonomy papers annually focusing on new species (~900) and the recent requirement for IJSEM to obtain genome sequences for new species. This argument is certainly valid. A somewhat less satisfying part of the response is that the GDBP method will reveal whether a query genome is monophyletic with an existing genus/species/subspecies and can therefore guide decisions on higher-level phylogeny. This is true, but the response mostly says that other methods need to be used.

General criticism 2: Utility of TYGS for less crowded parts of the tree of life.

The authors mostly refer users to 16S rRNA gene trees for this sort of task, which is certainly ok. Of course, conserved marker gene phylogenies and other approaches exist as well.

General criticism 3: The approach relies on the 16S rRNA gene, which may be absent or truncated in MAGs and SAGs.

The response here is that only MAGs and SAGs of high completeness and quality (and perhaps containing 16S rRNA genes) will be used for taxonomy in the future. Yes, this is true. This response emphasizes the intended use for taxonomists, rather than ecologists, which is already significant. A more important part of the response is that more than 50% of type strains currently don't have a genome sequence. This is true, but likely to change in the next two years and we will not be in this situation any longer.

Criticism 4: ANI and AAI would be good to add.

The authors respond that GBDP is superior to ANI since it corresponds more directly to DNA/DNA hybridization. This is true.

Criticism 5: Inclusion of non-type strains is limiting.

The authors stress that any user-specified genomes can be added.

Reviewers' comments

Reviewer #1

REVIEWER (Q1): Meier-Kolthoff and Goeker have resubmitted their manuscript "TYGS: an automated high-throughput platform for state-of-the-art genome-based taxonomy". Even though they claim that their manuscript was revised thoroughly, many points I raised in the first round of reviews have not been addressed.

RESPONSE (R1): The mere fact that we do not agree with certain proposals in the original review does not imply that we did not address all comments.

Indeed, some of the core concerns of the original review appear to be based on misunderstandings, which we addressed by re-writing and clarifying the according parts of the manuscript in our first revision. For instance, it is incorrect to postulate that we would need to conduct benchmarks for the genome-based species-delineation methods implemented in the TYGS, as these benchmarks were already published and cited in the manuscript and because better methods were not published in the meantime. We expect this to be quite obvious from the second, rephrased version of the manuscript and wonder why the benchmarking issue gets raised again without appropriately referring to our revised manuscript.

In fact, we have the impression that arguments from our rebuttal were not addressed by reviewer #1 in the second round of the revision and that the original criticisms were simply raised again. Details are given below.

REVIEWER (Q2): In very general terms I would like to comment on the authors rebuttal: 1) At least to me it seems to be unnecessary and bad style to copy-and-paste responses-to-reviewer-comments. At least I read the comments of other reviewers, both out of interest and in order to gather more ideas to help authors.

RESPONSE (R2): Within this submission system it is not obvious to the authors whether or not a particular reviewer is able to see the responses to the comments of another reviewer, and even if the reviewer could, separately providing equivalent responses to equivalent questions in distinct reviews is a completely reasonable approach. For this reason, this criticism of our rebuttal style seems to be entirely unnecessary.

The comment of reviewer #2 on this topic is revealing: "The format of your rebuttal was also much appreciated." There is some obvious redundancy in our responses but this redundancy can quite easily be explained by the redundancy in the original questions. The same holds for this round of the reviewing process, as much of the criticism can be traced back to few key issues.

REVIEWER (Q3): 2) It is generally accepted that not all reviewer comments are addressed and some can be discussed away within reason. It has been a while that authors have tried to discuss away the vast majority of my comments. As mentioned above, I am trying

top help the authors to improve their manuscript but it is also the duty of reviewers to maintain a good quality of scientific publications, so I implore the authors to take reviewer comments a bit more serious in general.

RESPONSE (R3): The mere fact that we do not agree with certain proposals in the original review does not imply that we did not take them serious.

As mentioned above, some of the core concerns of the original review appear to be based on misunderstandings, which we have addressed by re-writing and clarifying the according parts of the manuscript. For instance, it is incorrect to postulate that we would need to conduct benchmarks for the genome-based species-delineation methods implemented in the TYGS, as these benchmarks were already published and are cited in the manuscript and because better methods were not published in the meantime. In fact, the authors of the tools published in the meantime even failed to assess their methods against the relevant "ground truth". We expect this to be quite obvious from the second, rephrased version of the manuscript and had hoped for an appropriate reference to this revised manuscript in the case of additional disagreement.

Instead, we have the impression that arguments from our rebuttal were not taken serious by reviewer #1 in the second round of the revision and that the original criticisms were simply raised again. Further details are given below.

REVIEWER (Q4): First, I will go through the comments I previously made that have not been addressed properly yet. Afterwards, I will have some comments regarding the revision. I hope this will streamline the review process. Regarding Q2 [previous rebuttal]:

This was not only meant in terms of biological results, but as should be obvious from the phrasing of my comments this also pertains to the quality of the tool. Either the tool is shown to be useful (this includes the examples now listed, but these examples are neither benchmarks against a ground truth nor are they generally applicable) using a dedicated benchmark against competing tools (many of which have been published more recently tools in the TYGS workflow and hence could not possibly be compared before). What the authors have done here was to replace one example by two others. I hope my point it more clear now.

RESPONSE (R4): We nowhere claim that the examples we are providing can act as a replacement for a benchmark. Rather, we claim that benchmarks against the "ground truth" for bacterial species delineation (the traditional DDH) have already been conducted by us in previous publications and are appropriately cited in the manuscript, whereas better tools were not published since our original approach become available. It should be obvious from the already revised version of the manuscript that we reviewed the current literature on genome-based bacterial species delineation and did not find more recent publications that were benchmarked against the "ground truth", let alone yielded better results than the GGDC, whose species-delineation approach is incorporated in the TYGS. In fact, the more recent publications on bacterial genome-based species delineation did not even assess their methods against the "ground truth".

REVIEWER (Q5): Regarding Q4 [previous rebuttal]: The authors have ignored the main part of this comment, i.e. the term benchmark. No one knows how accurate TYGS is at the current point in time.

RESPONSE (R5): This claim is in disagreement with peer-reviewed, published and highly cited studies. The already published benchmarks for the GGDC species-delineation method reveal the accuracy of the species delimitation implemented in the TYGS simply because the two platforms use the same species-delineation method. Even if methods for bacterial species delineation that work at least as well as the GGDC were published in the meantime, one would still know how accurate the TYGS is.

It should be obvious from the already revised version of the manuscript that we reviewed the current literature on genome-based bacterial species delineation and did not find more recent publications that were benchmarked against the same "ground truth" (conventional DDH), let alone yielded better results than the GGDC, whose species-delineation approach is incorporated in the TYGS. In fact, the more recent publications on bacterial genome-based species delineation did not even assess their methods against the "ground truth".

REVIEWER (Q6): Regarding Q5 [previous rebuttal]: If this tool should be useful in a general sense, it should be possible to download a genome from NCBI and run it through TYGS. For this reason I asked the authors to "benchmark the recovery rates of SSU rRNA genes from incomplete genomes". The authors can claim that 16S is an essential gene for genomics but that doesn't make it magically appear within genomes. It is not my job to perform this task, but I have done this a few years ago and I was surprised how many genomes lack a 16S rRNA gene. I.e. I'm completely unsatisfied by the answer of the authors especially as my comments only pertained a quite feasible study that is highly relevant for TYGS.

RESPONSE (R6): This comment appears to be based on a misrepresentation of the views expressed in our manuscript. We nowhere "claim that 16S is an essential gene for genomics" but we claim that as yet the 16S rRNA gene is an essential gene for microbial taxonomy. This is a perfectly valid argument given the fact that the primary purpose of the TYGS is to assist taxonomists in genome-based species descriptions.

The second, revised version of the manuscript explains in detail why the 16S rRNA gene is still essential for bacterial taxonomy and classification. We argue that as long as a significant proportion of type strains lack a genome sequence the 16S rRNA gene remains essential for microbial taxonomy because in order to avoid the creation of younger heterotypic synonyms this nearly universally sampled gene must be examined. The reviewer may disagree with this view, but if so we had hoped for a counter-argument against our explanation in the second, revised version of the manuscript. Such a counter-argument is entirely missing; the reviewer simply re-raises the original criticism.

As should also be obvious from the second, revised version of the manuscript any bacterial genome sequence from NCBI could indeed be run through the TYGS; those that lack a 16S rRNA gene sequence (and thus should not be used for taxonomic purposes anyway!) are just a little bit more inconvenient to handle for the user (as they required a manual selection of reference genomes). Since for reasons that are obvious to taxonomists a significant proportion of genome sequences from NCBI would **not** be suitable for genome-based species descriptions anyway, it is **not** of relevance for the typical user of the TYGS that genome sequences lacking 16S rRNA genes are somehow less comfortable to handle.

For this reason, we do not think recovery rates of 16S rRNA genes in GenBank WGAs are of particular interest here. In particular, they have already been reported in the cited literature such as Yoon et al. 2017 (PMID: 28005526).

REVIEWER (Q7): Regarding Q6 [previous rebuttal]: I don't understand how my comment could be misunderstood but it has not been addressed. TYGS needs to be benchmarked, referring 5+ year old studies is not enough. (Also I don't understand in which context PMIDs: 19201692 is of relevance here). Adding random extra features does not address this comment either.

RESPONSE (R7): A scientific claim does not get disproved by age alone. The benchmarks against the "ground truth" for bacterial species delineation (i.e. the traditional DDH) that have already been conducted by us in previous publications and are appropriately cited in the manuscript may be some years old but this does by no means imply that better tools were published since then. In fact, it should be obvious from the already revised version of the manuscript that we reviewed the current literature on genome-based bacterial species delineation and did not find more recent publications that were benchmarked against the same "ground truth", let alone yielded better results than the GGDC, whose species-delineation approach is incorporated in the TYGS. In fact, the more recent publications on bacterial genome-based species delineation did not even assess their methods against the "ground truth". We thus did not misunderstand the original comment; we simply disagreed.

The reviewer further claims that we have incorporated "random extra features". This claim by the reviewer does not appear to be a scientific statement as it remains unexplained what these extra features are supposed to be, why they are supposed to be "random" with respect to the purpose of the TYGS, and why we are supposed to have claimed that we addressed any benchmark-related issues by mentioning these features.

We agree with the reviewer that instead of 19201692 the PMID 29088705 should have been provided. This does not affect the manuscript, however, in which the correct reference was used throughout.

REVIEWER (Q8): Regarding Q7 [previous rebuttal]: Again: benchmarking and comparing with other tools for species delineation! I used the search engine google scholar using the terms "species delineation protein coding marker genes" and successfully found examples and tools for this purpose. They might not be more accurate, but this is for the authors to show and prove.

RESPONSE (R8): To demonstrate that a certain method works better than a previously published method, the authors of the newer method must indeed compare their method against the previously published methods. Yet this raises the following question: Has any method been published after the publication of the species-delineation method implemented in the GGDC that—according to the established "ground truth" (conventional DDH)—works better than the method implemented in the GGDC?

The reviewer does not provide any actual examples, and the mere fact that one may obtain Google Scholar results for the query "species delineation protein coding marker genes" does not mean that these species delineation methods, if any, outperform the method used by the GGDC, or have been recommended by any taxonomic committee. If, however, none of the species-delineation methods published **after** the GGDC outperform the GGDC according to the established "ground truth" (conventional DDH), and if the TYGS implements the same species-delineation method as the GGDC (which it does),

then an additional benchmark study is apparently not necessary for the TYGS. Rather than asking whether we have compared the GGDC/TYGS species-delineation method against other methods (which we already did) one should ask whether the authors of species-delineation methods published **after** the GGDC did compare their method against the GGDC. If these authors failed to do so, this is not our mistake.

One wonders why it should be of any importance for our manuscript whether bacterial species delineation is feasible using "protein coding marker genes" as long as such marker-gene based methods were neither benchmarked against the ground truth (see above) nor demonstrated to outperform the species-delineation method of the GGDC. The mere fact that they were based on marker genes made these methods no better than the GGDC. Indeed, as selections of marker genes are not genomes, could methods based on such selections really be called genome-based taxonomy? This issue is also mentioned in our manuscript but not addressed by the reviewer at all.

The only relevant study found when searching for "species delineation protein coding marker genes" is the one by Mende et al. (2013) (PMID: 23892899) which was in fact published a month **after** the current GGDC 2.1 version (PMID: 23432962) but years **after** the initial GGDC 1.0 release (PMID: 21304684). Mende et al. (2013) would have had plenty of time to benchmark their approach against established methods as well the "ground truth" (conventional DDH).

The method implicitly suggested by the reviewer does not appear to be accepted in the taxonomic community as underlined by the relatively low number of Google Scholar citations (173) of Mende et al. (2013) compared to those of the GGDC 2.0 (1116), the GGDC 1.0 (629), the original average nucleotide identity (ANI) implementation (1042) and the slightly newer JSpecies ANI variant (1779). The method proposed by Mende et al. (2013) is not listed in the "Proposed minimal standards for the use of genome data for the taxonomy of prokaryotes" published in the IJSEM (PMID: 29292687).

While the "ad hoc committee for the re-evaluation of the species definition in bacteriology" (PMID: 12054223) recommended the evaluation of protein-coding gene sequence analysis as molecular criteria for species delineation they also mentioned that "In order to validate this approach, organisms should be chosen for which extensive DNA–DNA reassociation data are available [...]" (Stackebrandt et al. 2002; PMID: 12054223). Accordingly, much like any other bacterial species-delineation method, the method proposed by Mende et al. (2013) should have been evaluated against DDH data: "[...] Investigators are encouraged to propose new species based upon other genomic methods or techniques provided that they can demonstrate that, within the taxa studied, there is a sufficient degree of congruence between the technique used and DNA:DNA reassociation. [...]" (Stackebrandt et al. 2002; PMID: 12054223). This did not happen, however, as Mende et al. (2013) failed to consider the published recommendations.

Mende et al. (2013) compared the accuracy of their method with ANI but only analyzed selected clades. This raises the question how representative that subset still is to assess the overall accuracy of their tool. More importantly, ANI had been justified solely by its high correlation with conventional DDH. Since the according DDH data sets are publicly available, why did Mende et al. (2013) not validate their method using these data sets? The use of ANI instead only increases the level of indirection to DDH as already emphasized in our manuscript.

In fact, Mende et al. (2013) have chosen the least accurate ANI method for comparison, the MUMmer-based ANIm values (in addition to ANIb values), which have previously been shown to yield clearly worse correlation values compared to conventional DDH than other benchmarked ANI implementations did (PMID: 23432962), which were in turn outperformed by digital DDH method as implemented in the GGDC (PMID: 23432962).

Mende et al. (2013) used the NCBI Taxonomy database to assess the accuracy of their species classification method, although the NCBI taxonomy has never been authoritative (PMID: 22139910), thus likely affecting the outcome in one way or the other due to both misclassifications and misidentifications frequently present in that database. While the consideration of type strains by Mende et al. (2013) is an improvement on the reliance on the NCBI classification alone, many species with validly published names are known as heterotypic synonyms of each other. Accordingly, the current species classifications is certainly not to be regarded as the "ground truth" for optimizing species-delineation methods, particularly if the proper "ground truth" is easily accessible from the literature.

REVIEWER (Q9): I hope I could clarify my comments. At the current state I can't tell if this tool is better than the mentioned "GGDC web service". Also I am aware that DSMZ is an authority on the field of microbial taxonomy but as a reviewer I have to use the same standards that I hold everyone to. Hence, I cannot accept any argumentum ab auctoritate here. I hope that the authors take my comments a bit more serious this time and that the next version of their manuscript is more convincing!

RESPONSE (R9): Even though the species-delineation method of the TYGS is (for understandable reasons) the same as the one of the GGDC, the numerous additional features of the TYGS relative to the GGDC and the improvements for the user are so obvious and explained in detail in the manuscript that we see no reason to repeat them here.

The reviewer further claims that we rely on an "argumentum ab auctoritate". This claim by the reviewer does not appear to be a scientific statement as it remains unexplained which of our arguments is regarded as an "argumentum ab auctoritate" and for which reasons.

REVIEWER (Q10): New comments: a) The two new examples for delineating species need more outside evidence and discussion.

RESPONSE (R10): To us, this comment appears to be too unspecific to be of any help. It is not made obvious which additional kind of "outside evidence" would be expected. In particular, outside evidence is already provided in the manuscript for both the *Mycobacterium* and the *Salmonella* dataset. In either case we demonstrate that our results are new from a taxonomic viewpoint but not unexpected given certain earlier results using traditional methods and/or recently published results based on genome sequences.

REVIEWER (Q11): b) I tried the examples listed on the website and many genomes said something like: "Potential new species detected: your strain 'Bacillus anthracis str. Ames Ancestor' does not belong to any species found in TYGS database." I guess this is due to the limitation to type strains, but also shows a weakness (that another reviewer mentioned as well): Here it would be very useful to have non-type strains in order to connect with other researchers working on similar bacteria/genomes to decide on a type strain that can be explored in-depth later.

RESPONSE (R11): Taxonomy has to be based on nomenclature, and nomenclature is based on types. This is evident from the International Code of Nomenclature of Prokaryotes, which is appropriately cited in the manuscript. Disregarding the rules of the Code would cause microbial taxonomy to return to the anarchy of the period before the publication of the Approved Lists (Skerman et al. 1980).

As for the routine-inclusion of non-type strains in the TYGS database, of what use for the purpose of new genome-based species descriptions would be the comparison of a user-defined strain to a non-type strain? Since names of species are bound to their type strain, what would the comparison of some genome sequence to a non-type strain tell us? Whether or not they belong to the same species -- maybe. But which species would that be? This is impossible to tell as long the type strain is not considered.

In the case of a "potential new species detected" result the attentive TYGS user is directed towards conducting a 16S rRNA gene analysis with an extended set of type-strains, including type strains that were not yet genome-sequenced. As detailed in the manuscript, it is this kind of analysis that ensures a reliable taxonomic classification and nomenclature. To this end, switching to non-type strains would make matters only worse.

The solution to the *Bacillus anthracis* "mystery" is thus not to include non-type strains in the TYGS database but to sequence the genome of the type strain of *Bacillus anthracis*! (In this particular case a genome of potential *Bacillus anthracis* type strain is available on GenBank but since the exact origin of this strain has not been clarified yet, our automatic routines do not currently import this genome sequence. This is deliberate and neither a problem of the TYGS. Any alleged type-strain genome not yet contained in the database could be manually added by the user but it is then in the responsibility of the user to accept this as a type-strain genome.)

Reviewer #2

REVIEWER (Q12): Thank you for the clear and detailed responses to my previous comments. The format of your rebuttal was also much appreciated. I look forward to making use of the TYGS.

RESPONSE (R12): We are grateful for this positive evaluation.

Reviewer #3

REVIEWER (Q13): The authors have responded in detail to my criticisms and, in a few cases, made some improvements to the manuscript. I'm satisfied with their detailed responses, as discussed below. In the end, I think this database will be very important for taxonomists working on new species and subspecies of bacteria, but may not immediately be useful for a wider user community of ecologists. This is still a very significant impact.

RESPONSE (R13): We are grateful for this positive evaluation. We indeed plan to heavily support TYGS in the future and to add novel features, including those addressing topics in microbial ecology will of course be investigated.

REVIEWER (Q14): “General criticism 1: Focus on new species and subspecies rather than higher taxonomic ranks.” [original comment from first revision round]

The authors respond by stressing the large numbers of new taxonomy papers annually focusing on new species (~900) and the recent requirement for IJSEM to obtain genome sequences for new species. This argument is certainly valid. A somewhat less satisfying part of the response is that the GDBP method will reveal whether a query genome is monophyletic with an existing genus/species/subspecies and can therefore guide decisions on higher-level phylogeny. This is true, but the response mostly says that other methods need to be used.

RESPONSE (R14): Another method could be, however, a standard 16S rRNA gene analysis, which in the case of less densely sampled groups can well yield sufficient phylogenetic resolution. One of the advantages of the TYGS is that this standard 16S rRNA gene analysis can directly be triggered by the user and will then contain all type strains of taxonomic interest. Thus one of the other methods is directly available in the TYGS, and the integration of these methods with our up-to-date nomenclature database guarantees that no closely related type strain will not be overlooked. In more densely sampled groups the GDBP tree may already contain all information that is needed to arrive at taxonomic conclusions.

REVIEWER (Q15): “General criticism 2: Utility of TYGS for less crowded parts of the tree of life.” [original comment from first revision round]

The authors mostly refer users to 16S rRNA gene trees for this sort of task, which is certainly ok. Of course, conserved marker gene phylogenies and other approaches exist as well.

RESPONSE (R15): We fully agree, but the coverage of type strains by such conserved marker genes is significantly lower than their coverage by 16S rRNA gene sequences. As we have emphasized particularly in the revised version of the manuscript, all closely related type strains must have been compared in order to safely propose a new species and to avoid the creation of a heterotypic synonym. For this reason, we cannot currently dispense with the 16S rRNA gene for taxonomic purpose although it would be advantageous if we could.

REVIEWER (Q16): “General criticism 3: The approach relies on the 16S rRNA gene, which may be absent or truncated in MAGs and SAGs.” [original comment from first revision round]

The response here is that only MAGs and SAGs of high completeness and quality (and perhaps containing 16S rRNA genes) will be used for taxonomy in the future. Yes, this is true. This response emphasizes the intended use for taxonomists, rather than ecologists, which is already significant. A more important part of the response is that more than 50% of type strains currently don't have a genome sequence. This is true, but likely to change in the next two years and we will not be in this situation any longer.

RESPONSE (R16): It is difficult to predict when the point in time at which we can dispense with the 16S rRNA gene for taxonomic purposes will actually be reached. While journals such as IJSEM strongly recommend the genome sequencing of type strains used in species descriptions, this is still not mandatory, hence new species without type-strain genomes are still proposed in the literature. Moreover, obtaining the genome sequences

for all the older validly published species and subspecies is a tedious task that requires the collaboration between several culture collections and sequencing centers. We are heavily involved in type-strain genome sequencing projects ourselves since 2008 and believe two years to be a too optimistic estimate. In the course of the GEBA-VI project that was approved in 2018 by the DOE-JGI, the DSMZ was asked to provide DNA for the genome sequencing of 5,000 type strains deposited in our collection. Even if we were able to provide 1,000 DNAs per year we would need 5 years to complete this task, but 1,000 DNAs per year for a single large-scale project is currently beyond the cultivation and DNA-preparation capability of the DSMZ and also beyond the sequencing capability of the JGI. Our latest count yielded 7,500 type strains of species or subspecies whose genome has not yet been sequenced; significant proportions of these type strains are not even deposited in the DSMZ collection.

REVIEWER (Q17): “Criticism 4: ANI and AAI would be good to add.” [original comment from first revision round]

The authors respond that GBDP is superior to ANI since it corresponds more directly to DNA/DNA hybridization. This is true.

RESPONSE (R17): We are grateful for this positive response.

REVIEWER (Q18): “Criticism 5: Inclusion of non-type strains is limiting.” [original comment from first revision round]

The authors stress that any user-specified genomes can be added.

RESPONSE (R18): We are grateful for this positive response.